# Attributing near-surface atmospheric trends in the Fram Strait region to regional sea ice conditions

Amelie U. Schmitt[1,2] and Christof Lüpkes[3]

[1]Meteorological Institute, Center for Earth System Research and Sustainability (CEN), Universität Hamburg, Hamburg, Germany
[2]now at Climate Service Center Germany (GERICS), Helmholtz-Zentrum Hereon, Hamburg, Germany
[3]Alfred Wegener Institute, Helmholtz Centre for Polar and Marine Research, Bremerhaven, Germany

**Correspondence:** Amelie Schmitt (amelie.schmitt@hereon.de), Christof Lüpkes (christof.luepkes@awi.de)

**Abstract.** Arctic sea ice has declined in all seasons accompanied by rapid atmospheric warming. Here, the focus lies on the wider Fram Strait region where the connection between trends in observed near-surface variables (temperature, humidity, wind speed) and local sea ice conditions are analyzed. Reanalysis data from ERA5 and MERRA-2 and SSM/I-ASI sea ice concentrations for the winters of 1992 to 2022 are used for the analyses.

Two focus regions are identified for which trends are largest. In the Western Nansen Basin (WNB), sea ice cover decreased by -10 % $\mathrm{dec}^{-1}$ with especially large open water areas in 2022, and temperature and humidity increased by up to $3.7\,\mathrm{K}$ and $0.29\,\mathrm{g\,kg}^{-1}$ per decade, respectively. In the Greenland sea region (GRL), trends were slightly smaller with -4.7 % $\mathrm{dec}^{-1}$ for sea ice and up to $1.3\,\mathrm{K}$ and $0.15\,\mathrm{g\,kg}^{-1}$ per decade for temperature and humidity. Trends for wind speed were mostly not significant.

As a next step, two typical flow directions for this region were studied: cold-air outbreaks with northerly winds originating from ice covered areas (off-ice flow) and warm-air intrusions with southerly winds from open ocean regions (on-ice flow). To identify possible relationships between sea ice changes and atmospheric trends, correlation maps were calculated and the results for off- and on-ice flow were compared. Up to two thirds of the observed temperature and humidity variability in both regions are related to upstream sea ice variability and an influence of sea ice cover is still present up to $500\,\mathrm{km}$ downstream of the ice edge. In the marginal sea ice zone the impact of a decreasing sea ice cover in this region is largest for off-ice flow conditions during cold air outbreaks.

## 1   Introduction

In the last decades the Arctic climate changed rapidly with a much stronger increase of atmospheric temperatures than in mid latitudes. This phenomenon, called the Arctic Amplification, has been described in many articles (e.g. Cohen et al., 2014; 20  Graversen et al., 2008). Temperature increase went along with an unprecedented decrease of sea ice extent (SIE) (Stroeve et al., 2012, 2014). Shu et al. (2020) analyzed different satellite datasets from 1979 to 2014 and found that SIE decreased in all seasons with trends of about -0.82 and -0.35 million km$^2$ per decade in September and March, respectively. The strength of the trend also highly depends on the region. Stroeve and Notz (2018) and Onarheim et al. (2018) found that the regions with

the largest decrease of SIE were the Beaufort and East Siberian Seas in September and in the Barents and Greenland Seas in
March.

In this study, we focus on winter conditions in the Fram Strait region, which contains parts of the Greenland and Barents Seas. Our aim is to identify ongoing changes of regional atmospheric near-surface conditions and their relations to observed changes in sea ice cover. Various studies have already addressed the close connection of sea ice loss with processes in the atmosphere and ocean. For example, sea ice changes can influence atmospheric variables such as the regional pressure field (Schneider et al.,
2021) or cloud radiative forcing (Stapf et al., 2020). Decreasing sea ice concentration also impacts atmospheric temperature, wind and density stratification in the atmospheric boundary layer (e.g. Lüpkes et al., 2008; Tetzlaff et al., 2015; Chechin et al., 2019; Michaelis et al., 2021; Wang et al., 2019).

One area of specific interest for regional studies has been the Greenland Sea. Moore et al. (2015) showed that the wintertime retreat of sea ice in the Greenland and Iceland seas reduced the magnitude of air-sea fluxes by 20 % since 1979 with potential
future implications for the Atlantic Meridional Overturning Circulation. Selyuzhenok et al. (2020) found a link between the decrease of sea ice volume in the Greenland Sea between 1979 and 2016 and an increase of the amount of upper-ocean heat content available for sea ice melt related to increased temperatures of the Atlantic Water inflow in this region. An interesting local phenomenon in this region is the so-called Odden ice tongue, which occurred regularly until the early 1990ies in the region influenced by the Jan Mayen current (Wadhams and Wilkinson, 1999). Comiso et al. (2001) showed that the extent of
the Odden ice tongue between 1979 and 1998 was affected by surface winds and had a strong negative correlation with the monthly surface air temperature recorded at Jan Mayen island.

Several other studies also considered the region around Svalbard and proposed different mechanisms that could explain recent sea ice trends. Ivanov et al. (2012) concentrated on the role of Atlantic water inflow in shaping the ice conditions between Svalbard and Franz Joseph Land. They found that sea ice conditions around Svalbard substantially affect the temperature
regime of the Spitsbergen archipelago, particularly in winter. They conclude that in the Atlantic Water inflow, heat flux from the ocean reduces the sea ice thickness and thus trends in the ocean currents around Svalbard could have a profound effect on Svalbard climate. Dahlke et al. (2020) analyzed the sea ice variability around Svalbard from 1980 to 2016 and found a different explanation for decreasing SIE in the northern fjords in winter in the last 15-20 years. They calculated lag correlations with a time shift of 1 month, which indicate that atmospheric warming leads the sea ice signal and not vice versa. Lundesgaard et al.
(2021) considered the water inflow region north of Svalbard and state that the sea ice concentration in this region is influenced by many different factors including atmospheric and oceanic forcing. They find, e.g. that between 2012 and 2019, heat flux from the ocean could not explain the inter-annual variability in sea ice concentration but sea ice drift was playing a key role. Isaksen et al. (2016) relate the recent strong warming of Spitsbergen to variations in the large scale atmospheric circulation, air mass characteristics and sea ice concentration around Spitsbergen. They find a high correlation between the land-based surface
air temperature and local and regional sea ice concentration and finally that the strong warming in recent decades over the Svalbard Archipelago is driven by higher sea surface temperature and sea ice decline.

In this study, we consider changes in meteorological quantities and sea ice cover in two regions, which are the region north of Spitsbergen and the region along the eastern Greenland coast. The first one was in the center of two studies, one by Onarheim

et al. (2014) and another one by Tetzlaff et al. (2014), which we refer to as Tetz14 in the following . Both showed a strong sea ice decline in the region of the Whalers Bay polynya at the northern coast of Spitsbergen. Onarheim et al. (2014) shows that in this region loss of sea ice concerns especially winter, while in the entire Arctic Ocean sea ice is shrinking mainly during summer. Tetz14 find a close connection with local atmospheric processes such as an unprecedented high value of the atmospheric convective boundary-layer height in cold-air outbreaks during spring 2013 caused by the northward shift of the ice edge.

Besides focusing on sea ice trends, we also analyze regional trends of near-surface air temperatures and - departing from previous studies - also of humidity and wind speed between January and March for the years 1992 to 2022. Our results are based on data from the ERA5 and MERRA-2 reanalyses, which have a high spatial resolution and are thus ideal for studying also smaller regional differences. We then aim to answer the following questions:

1. What are the trends of sea ice concentrations and near surface atmospheric variables in the Fram Strait region? (Sect. 3.1)

2. Two typical synoptic situations in this region are cold air outbreaks (CAOs) with cold, dry air flowing from sea ice covered regions towards the open ocean and warm-air intrusions bringing in warm and moist air during southerly flows. What atmospheric trends can we observe for these two situations – off-ice flow during CAOs and on-ice flow during warm-air intrusions? (Sect. 3.2)

3. One factor influencing the strength of CAOs in this region is the open water fetch or the fraction of ocean covered by sea ice. Can we find a relationship between the atmospheric trends over the open ocean in the Fram Strait region and a decrease of sea ice in the upstream region? We address this question by calculating correlation coefficients between sea ice cover and atmospheric quantities and then separately analyzing results for off- and on-ice flow conditions. The idea behind this is that especially during off-ice flow strong convection occurs over the open ocean, and the highest vertical fluxes of heat and humidity occur near the ice margins where they have a profound impact on processes in the atmospheric boundary layer. On the other hand, warm-air intrusions transport warm and humid air masses northward, so that there are two competing mechanisms related to on- and off-ice flow. Thus - even though we cannot completely exclude additional impacts by large scale atmospheric processes or changed ocean currents - comparing correlations for different wind direction sectors helps us to identify regions where sea ice retreat is very likely responsible for atmospheric trends. (Sect. 3.3)

The used datasets and applied methods are described in Sect. 2, followed by a presentation and discussion of the results in Sect. 3. A summary and conclusions are presented in Sect. 4.

## 2 Data and methods

### 2.1 Sea ice concentration

To analyze sea ice trends in the Fram Strait region we use the SSM/I-ASI sea ice concentration dataset provided by Ifremer. Sea ice concentration is derived from space-borne passive microwave measurements – namely brightness temperatures measured at the 85 GHz channel of the Special Sensor Microwave / Imager (SSM/I) and its successor, the Special Sensor Microwave / Imager Sounder (SSM/IS). The ARTIST Sea Ice (ASI) algorithm (Kaleschke et al., 2001; Spreen et al., 2008) is applied to compute sea ice concentration, followed by a 5-day median filter to reduce the weather impact (Kern et al., 2010). The data are provided on a polar-stereographic grid with a resolution of 12.5 km x 12.5 km. In this study, we use 31 years of winter data (January – March) from 1992 to 2022.

### 2.2 Reanalysis datasets

The fifth generation of the ECMWF atmospheric reanalysis (ERA5) provides hourly data with a horizontal resolution of $0.25° \times 0.25°$ on 137 vertical levels (Hersbach et al., 2020). Here, we use air temperature, specific humidity and horizontal wind components from the lowest model level, which has an average height of about 9 m for this region and season. Data for the years 1992 to 2022 were downloaded from the Copernicus Climate Change Service (C3S) Climate Data Store (Hersbach et al., 2017).

Version 2 of the Modern-Era Retrospective analysis for Research and Applications (MERRA-2) provides hourly data on a $0.5°$-latitude and $0.625°$-longitude grid and on 72 vertical levels (Gelaro et al., 2017). Here, we use air temperature, specific humidity and horizontal wind components at 10 m height for the years 1992 to 2022 from the single level diagnostics dataset provided by Global Modeling and Assimilation Office (GMAO) (2015).

### 2.3 Study region and WNB fetch length

We study changes of sea ice and atmospheric conditions in the wider Fram Strait region. Based on the trends presented in Sect. 3.1 we select two specific focus areas (Fig. 1). The first one is located at the Western Nansen Basin (WNB) north-east of Svalbard - also called the Whaler's Bay polynya - which has been the study area in previous works by Ivanov et al. (2012), Onarheim et al. (2014) and Tetz14. The second focus area is located in the Greenland Sea (GRL) at the east coast of Greenland.

We specify areas for which we calculate sea ice trends (ICE boxes), which are presented in Sect. 3.1 (see Fig. 1). For the WNB region we place the ICE box in a similar location as used in previous studies by Ivanov et al. (2012) and Tetz14. Our aim was to place the GRL ICE box in such a position that it covers most of the extent of the Odden ice tongue, which is present in some of the considered years, and thus we chose the location of the box accordingly. This means, however, that the GRL ICE box contains much more open ocean regions than the WNB ICE box and thus the sea ice cover trends are expected to be smaller.

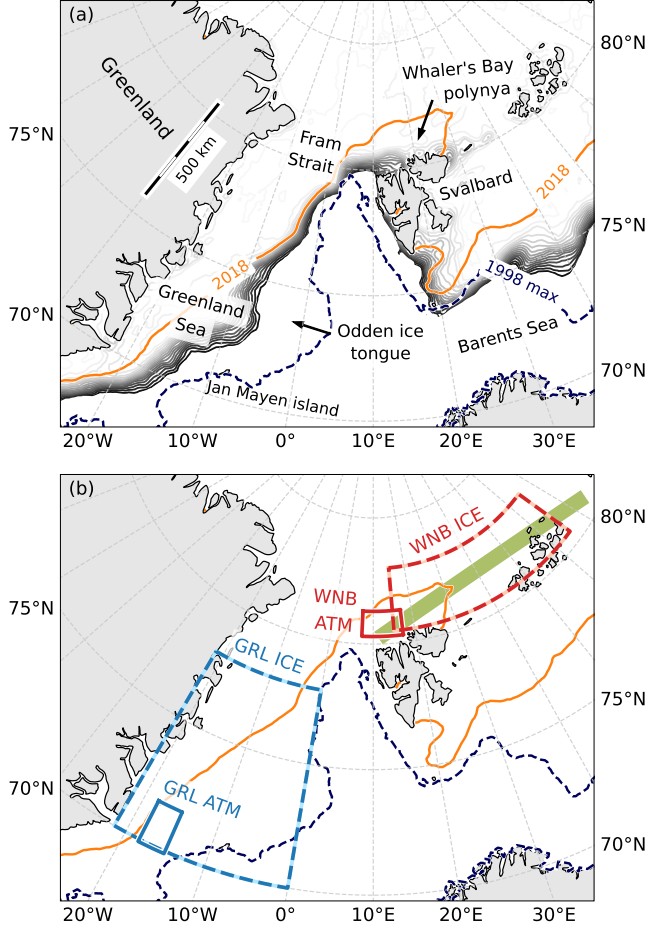

**Figure 1.** Overview maps of the study region. (a) January to March averages of the SSM/I-ASI sea ice concentration are displayed as grey contours from 15 % upward in 5%-steps for 1996 (a year with large SIE) and as orange 15 %-contour for 2018 (a year with small SIE). The 15 %-contour of the maximum sea ice concentration in 1998 (a year with a pronounced Odden ice tongue) is shown as dashed line. (b) Location of the specific study areas: Large ICE boxes indicate the areas considered for trends of sea ice conditions and small ATM boxes indicate areas considered for atmospheric trends in the Greenland Sea (GRL, blue) and the Western Nansen Basin (WNB, red) regions, respectively. The thick green line indicates the path considered for WNB fetch calculations.

We also study trends of atmospheric variables in a region influenced by the changing sea ice cover in the WNB and GRL regions. Besides focusing on regional patterns within and downwind of the ICE boxes, we also conduct time series analyses in two smaller ATM boxes. More details about how the locations of the ATM boxes were chosen can be found in Sect. 3.1. The exact location of all boxes considered for trends in sea ice (ICE box) and atmospheric variables (ATM box) can be found in Tab. 1.

Tetz14 determined the distance an air parcel traveled over open ocean in the WNB region and showed that air temperatures downstream of this region correlate with this so-called fetch length. Here, we also calculate the WNB fetch length along the thick green line indicated in Fig. 1b and consider grid cells with less than 70 % sea ice concentration as open water (see Sect. 3.3).

**Table 1.** Coordinates of the Western Nansen Basin (WNB) and Greenland Sea (GRL) boxes considered for the statistical analyses of sea ice (ICE) and atmospheric (ATM) conditions (see also Fig. 1).

| Region | Latitude | Longitude |
|---|---|---|
| **WNB** | | |
| ICE box | 80.5 - 83.0 °N | 15.0 - 65.0 °E |
| ATM box | 80.3 - 81.3 °N | 7.3 - 17.1 °E |
| **GRL** | | |
| ICE box | 70.0 - 78.0 °N | 21.5 - 0.0 °W |
| ATM box | 69.9 - 71.7 °N | 18.2 - 14.9 °W |

## 2.4 Statistical methods

We calculate the slopes $b$ of all trend lines using a linear least-squares regression. The uncertainty of the slopes is given as the 95 %-confidence interval according to a t-test. Using the t-value $t$ for $n - 2$ degrees of freedom (with $n = 31$ years) and the standard error $s_b$ of the slope, the confidence interval is calculated as $\pm s_b\, t_{n-2}$. To determine whether two different trends (e.g. for off- and on-ice flow) differ significantly from each other we calculate a t-test of the slopes:

$$t = \frac{b_1 - b_2}{s_{b1}^2 + s_{b2}^2},$$

(1)

with $df = n_1 + n_2 - 4$ degrees of freedom.

Correlations are calculated to determine the relationship between sea ice and meteorological variables. Since the relationship is non-linear (see Fig. 12), we use the Spearman's rank correlation coefficient $r$, which is a measure of the strength of a monotonic relationship. A t-test is applied for significance testing of correlation coefficients. To test whether two different correlation coefficients differ significantly from each other, we use the formula for independent correlation coefficients (Steiger, 1980), which is based on Fisher's z-transform:

$$z = \frac{1}{2} \ln\left(\frac{1+r}{1-r}\right).$$

(2)

A t-test is then applied to the difference of the $z$-values.

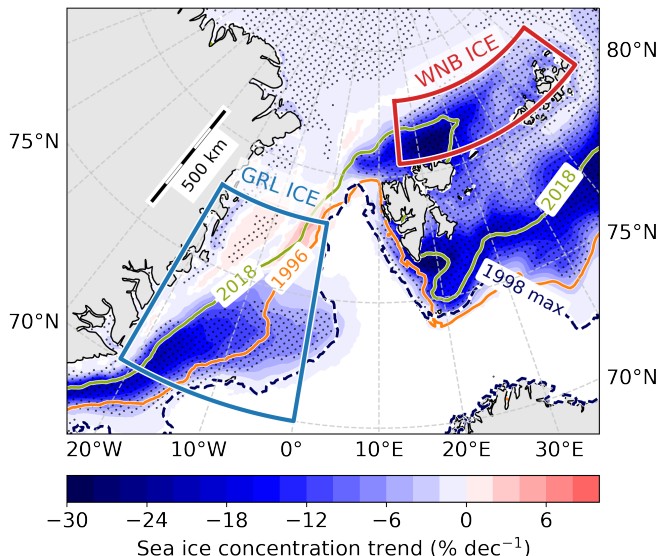

**Figure 2.** Sea ice concentration trends based on SSM/I-ASI data for the years 1992 to 2022. Dotted areas are significant at the 95 %-level. Colored lines denote the 80 % SSM/I-ASI sea ice concentration contours averaged from January to March for two years with large and small SIE, respectively. The dashed contour shows the maximum SIE between January and March in 1998, which roughly represents a typical outline of the Odden ice tongue. ICE boxes used for the calculation of sea ice concentration time series are shown in blue for the GRL region and in red for the WNB region.

## 3 Results and discussion

In this section we first present trends in atmospheric variables and sea ice concentrations in the wider Fram Strait region (Sect. 3.1). We then choose two focus regions (WNB and GRL) based on the locations of the largest trends in the wider region, calculate atmospheric trends separately for off- and on-ice flow directions and compare the results (Sect. 3.2). Finally, to assess the impact of the observed sea ice variability on atmospheric conditions, we calculate correlation maps (Sect. 3.3) and discuss possible mechanisms for these correlations in Sect. 3.4.

### 3.1 Trends in the Fram Strait region

Using data from two reanalyses and a sea ice concentration dataset from 1992 to 2022, we calculate long-term trends in the Fram Strait region. Trends of sea ice concentration based on SSM/I-ASI data indicate that in the 31 years since 1992 sea ice cover has declined in large parts of the Fram Strait region (Fig. 2). The largest trends exceeding -20 % per decade are observed in the Barents Sea south-east of Svalbard, in the region of the Whaler's Bay polynya north of Svalbard and in the region of the Odden ice tongue in the Greenland Sea. Based on these results, we choose the region of the Whaler's Bay polynya/Western Nansen Basin (WNB) and the Greenland Sea region (GRL) as the two main focus areas for the analyses in this study (see also Sect. 2.3).

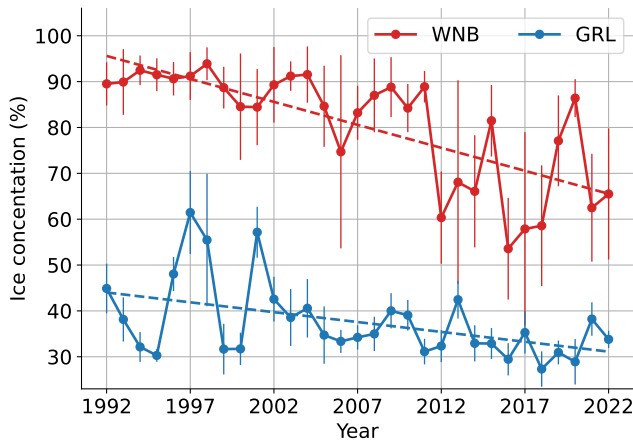

**Figure 3.** Winter mean sea ice concentrations from SSM/I-ASI in the ICE boxes in Fig. 1b and 2 for the WNB and GRL regions averaged from January to March. Errorbars denote standard deviations of daily values. Dashed lines represent linear trends.

We calculate average ice concentrations within the ICE boxes indicated in Fig. 1b and Fig. 2 for the months January to March.
Both the WNB and the GRL regions exhibit decreasing trends since 1992 (Fig. 3). In the WNB region the trend is -10.0 ± 3.5 % per decade with changing variability over time. In the 1990s, inter-annual variability was small with almost constant average sea ice concentrations around 90 %. Between 2000 and 2011 the variability increased and sea ice concentration approached 80 %. After 2012, we observe another shift in mean sea ice conditions with a strong inter-annual variability. In the GRL region the trend is also negative with -4.3 ± 3.1 % per decade (Fig. 3). It is smaller than in the WNB region, which is partly because
the GRL ICE box contains open ocean areas and thus the maximum observed average sea ice concentration is more than 10 % lower than for the WNB region.

These results are in line with sea ice trends in the winter seasons reported by previous studies. Based on NASA-Team passive microwave sea ice concentration data from 1979 to 2018, Stroeve and Notz (2018) found trends of the same order of magnitude in both the WNB and GRL regions for March. Wang et al. (2019) considered a slightly larger region in the Greenland Sea,
which extended further south and thus contained more open ocean areas. Consequently, they found a slightly smaller trend of about -2 % per decade in winter from 1982 to 2016.

Like in the WNB region, the inter-annual variability also changes in the GRL region during the considered period. There, larger variability is observed in the first half of the period while it is vice versa in the WNB region, as already mentioned. One possible explanation for the larger variability in the GRL region is the presence of the Odden ice tongue, which increases
the average ice concentration in this region. From Fig. A1 it is evident that for example 1997 and 1998 are years with a large Odden ice tongue and also two of the years with the overall largest observed mean sea ice concentrations since 1992. After 2006 the Odden ice tongue does not occur frequently any more and thus the inter-annual sea ice variability decreases. Sumata et al. (2023) analyzed the sea ice thickness distribution in the Arctic and found a regime shift from thicker and deformed to

thinner and more uniform ice cover around the year 2007. This timing also coincides with the change in inter-annual variability

of the sea ice concentration in the WNB region, where standard deviations of 4.6 % from 1992 to 2006 where much lower than the values of 12.3 % observed from 2007 to 2022.

We also calculated trends of atmospheric variables in the Fram Strait region based on reanalysis data (Fig. 4). Trend maps for near-surface air temperatures and specific humidity look very similar for the two different reanalyses ERA5 and MERRA-2. Temperature trends are generally positive and they are the largest north-east and east of Svalbard (Fig. 4a,b) roughly in the

same regions where we also observe the largest sea ice reductions over the 31-year period (see Fig. 2). The maximum warming of about 3.7 K per decade for ERA5 and 3.4 K per decade for MERRA-2 can be found in the region of the Whaler's Bay polynya (see Fig. 1a for region names). This coincides with the region of the maximum humidity trends of $0.27\,\mathrm{g\,kg^{-1}}$ per decade for ERA5 and $0.29\,\mathrm{g\,kg^{-1}}$ per decade for MERRA-2. A second area with locally increased trends is located in the southern Greenland Sea region where observed trends are, however, less than half as large as in the Whaler's Bay region. These

maxima also coincide with the area of largest sea ice decreases in the Greenland Sea region. Trends at the northern coast of Norway and the north-eastern coast of Greenland are smaller and not significant at the 95%-level.

Trends for wind speed are generally smaller and less significant than for temperature and humidity. Also, there are a few notable differences between the two reanalysis datasets (Fig. 4e,f). The only significant increase of wind speed over the 31-year period can be found in the region of the Whaler's Bay polynya for ERA5 data, while MERRA-2 trends in this region are close

to zero. Both reanalyses show the largest negative trends at the east coast of Greenland but only for MERRA-2 significant negative trends also extend further across the Fram Strait towards the coast of Svalbard.

Temperature trends for similar regions have previously been calculated by other authors. Johannessen et al. (2016) analyzed the Nansen surface air temperature dataset, which incorporates data from land stations, buoys and ship measurements in a larger region containing the Greenland and Barents Seas and the Fram Strait region. They found that running 30-year temperature

trends increased since the 1970s with a value exceeding 1.5 K per decade from 1985 to 2014. Even though they considered a larger region, our results for WNB indicate that temperature trends might have further increased since 2014. Based on air temperature data from the NCEP-DOE reanalysis from 1979 to 2016, Wang et al. (2019) reported trends of about 1.5 K per decade in the GRL region and of about 2 K at the north-west corner of Svalbard, which is in line with our findings. In a recent study, Isaksen et al. (2022) presented air temperature trends based on weather station and reanalysis data averaged from

December to February. Between 1991 and 2020, trends in the region north-east of Svalbard were in the order of 3 K per decade based on ERA5 data and even slightly larger values based on station data and the CARRA reanalysis, which is also in line with our findings.

As a next step, we analyze wind roses for selected sub-regions, focusing especially on those regions where observed trends for sea ice cover and air temperature and humidity were largest - namely the WNB and GRL regions. We calculate the wind

roses for smaller sub-regions, which we refer to as ATM boxes. The exact placement of these boxes is somewhat arbitrary and could be based on many different factors. We decided to choose locations where observed temperature and humidity trends in Fig. 4 are large. Furthermore, we choose areas that were only covered by sea ice in years with a large ice extent but were mostly ice-free in years with a small sea ice cover. Due to the large change in local ice cover we also expect to see a notable

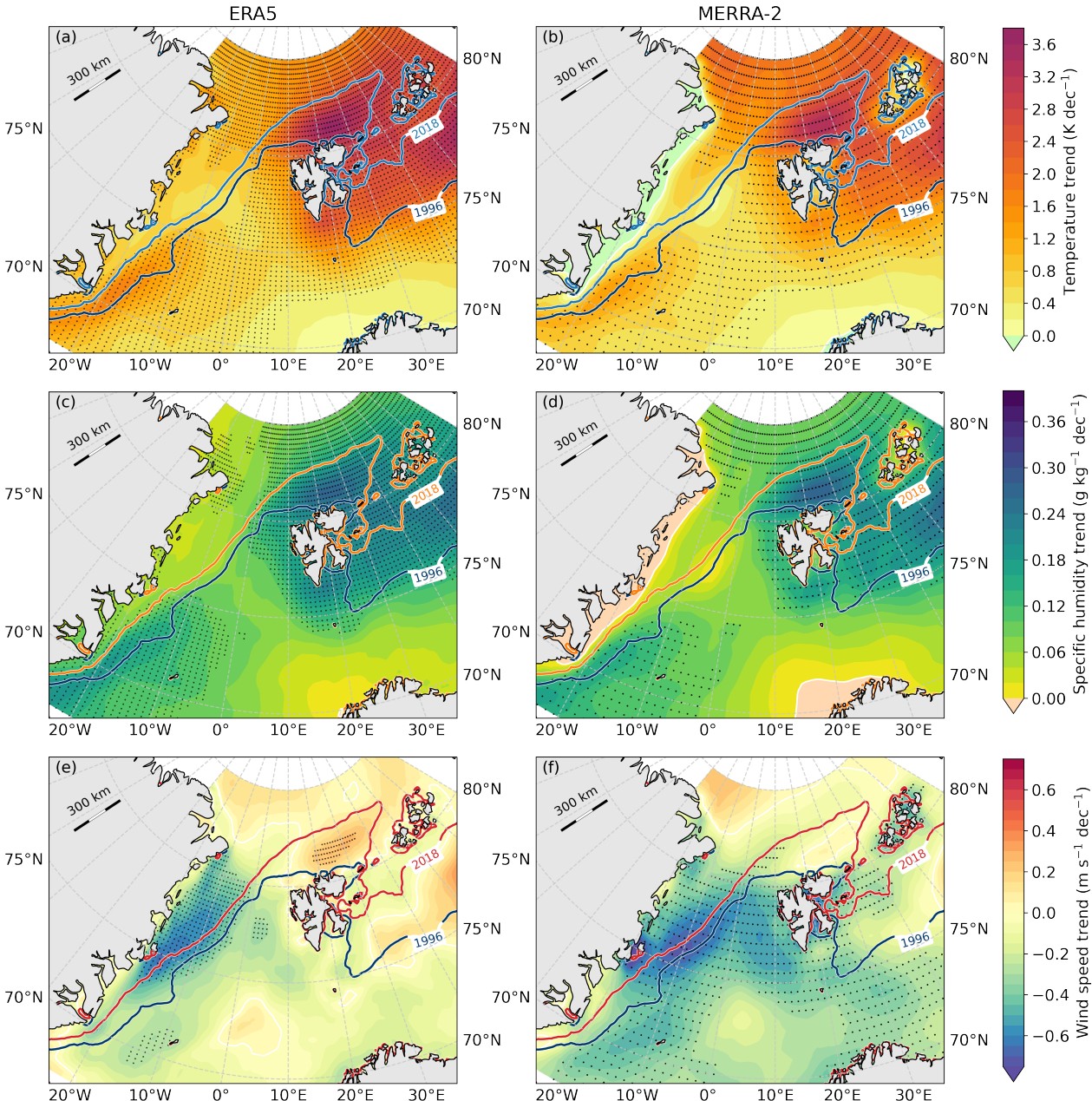

**Figure 4.** Trend maps for atmospheric variables in the Fram Strait region based on ERA5 (left) and MERRA-2 (right) reanalysis data from January to March of the years 1992 to 2022. Trends are shown for air temperature (a,b), specific humidity (c,d) and wind speed (e,f). Dotted areas are significant at the 95 %-level. Blue lines denote the 80 % SSM/I-ASI sea ice concentration contours averaged from January to March for two years with large and small SIE, respectively.

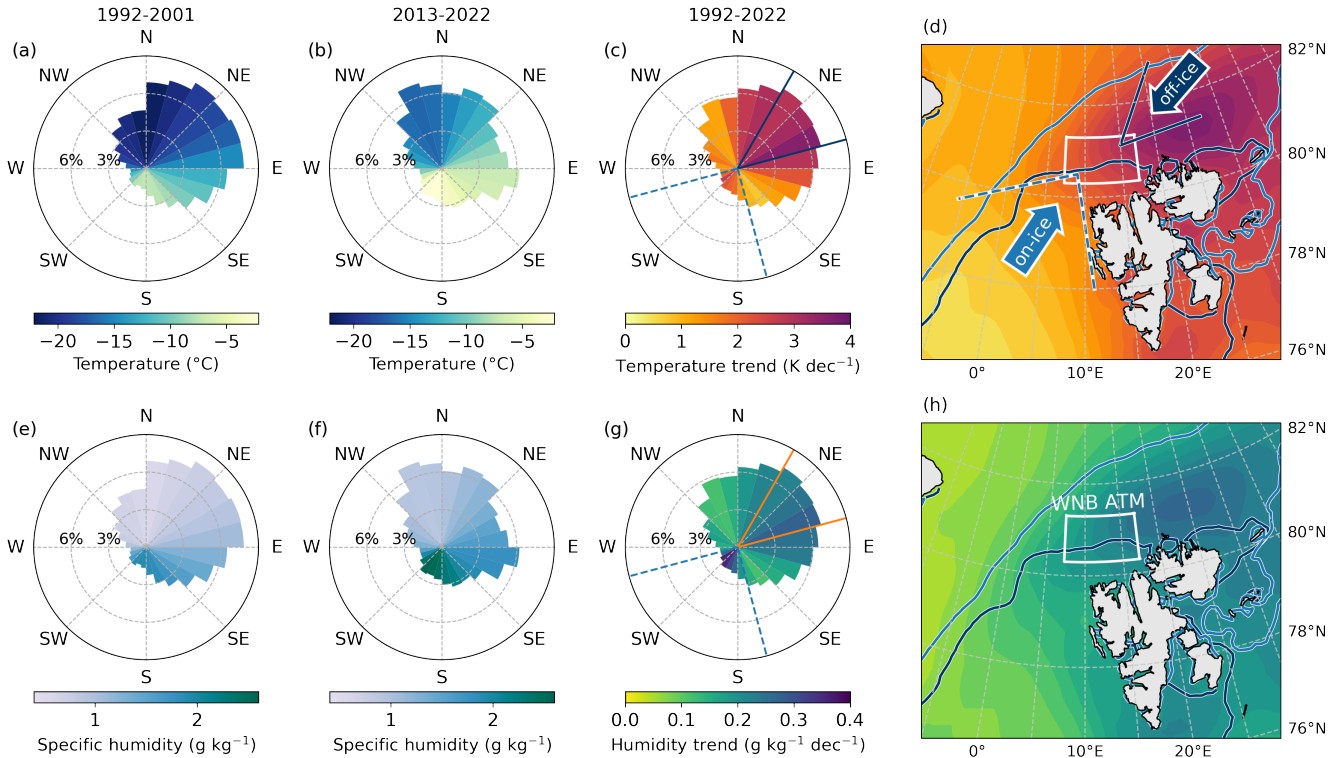

**Figure 5.** Wind roses based on ERA5 data in the WNB ATM box for the first ten years of the study period (a,e), the last ten years (b,f), and the whole period from 1992 to 2022. The length of the bars indicates the observed frequency of winds from each wind direction sector spanning 15°. The color denotes the average air temperature (a,b) and specific humidity (e,f) for the respective periods. The colors in panels (c) and (g) show the corresponding temperature and humidity trends over the whole 31 years. Blue and orange lines denote the sectors considered for off- and on-ice flow. Maps illustrating the locations of the WNB ATM box and the flow directions for off- and on-ice flow are shown with trends for all wind directions of temperature (d) and specific humidity (h) as background shading. Blue lines denote the 80 % SSM/I-ASI sea ice concentration contours averaged from January to March for two years with large (1996) and small (2018) SIE, respectively.

impact on atmospheric conditions in these areas. For the exact locations of the WNB and GRL ATM boxes see Fig. 5d,h and 6d,h and Tab. 1. Results based on ERA5 and MERRA-2 data show only small differences, and thus we only present and discuss ERA5 results in the following. In the WNB ATM box, winds originate mostly from northerly and north-easterly directions. Air masses originating from north are the coldest and driest (Fig. 5a,b,e,f) and air masses originating from the south and south-west are warmer and more humid. Wind directions in the GRL ATM box show much less variability than in the WNB ATM box with winds coming most often from northerly directions (about 40 % of the time). Similar to the WNB region the coldest and driest air masses originate from the north and north-west while air masses from the south-east are warmer and more humid (Fig. 6a,b,e,f).

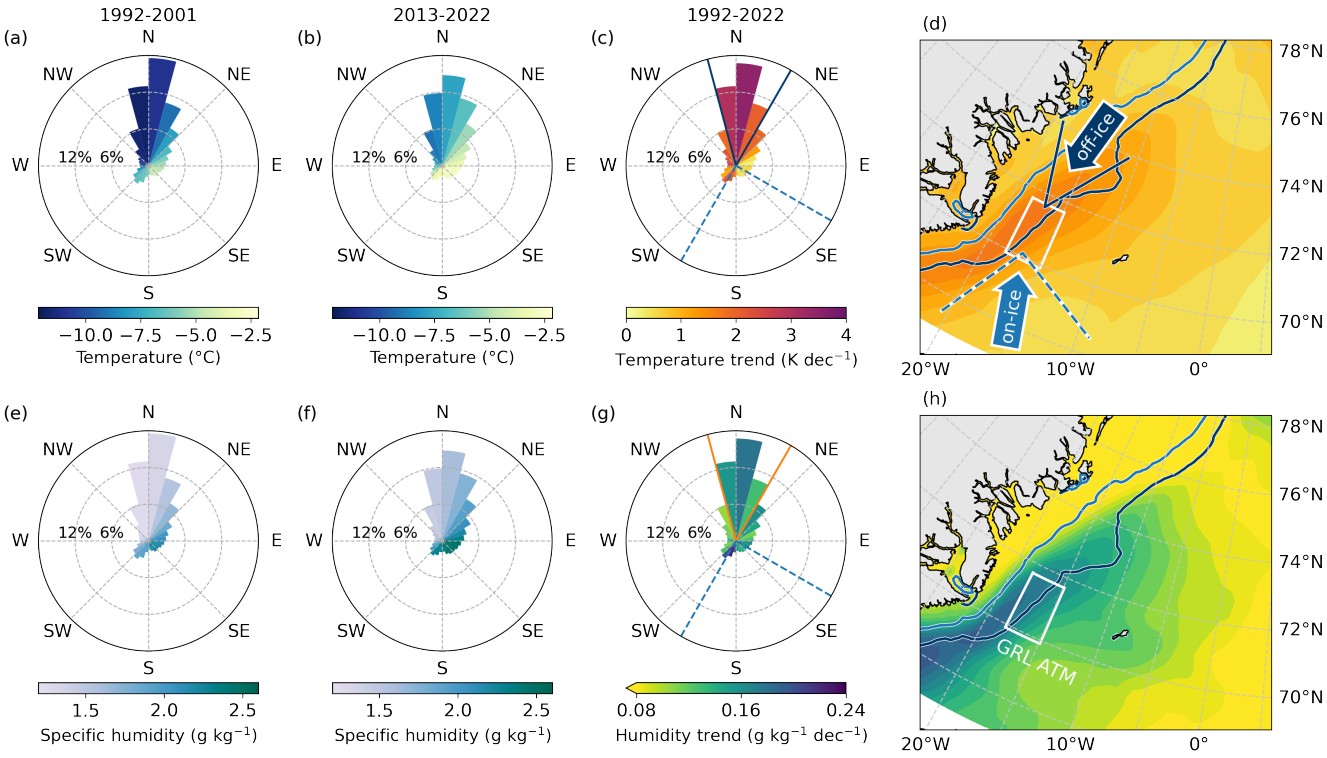

**Figure 6.** Same as Fig. 5 but for the GRL region. Note that the colormap of panels (g) and (h) spans a smaller range than in Fig. 5.

## 3.2 Atmospheric trends for off-ice and on-ice flow

The analysis of wind roses in the previous section also enables us to analyze whether the trends observed in Fig. 4 are stronger for certain wind directions. In the WNB ATM box, temperature trends of about 3.5 K per decade are observed for north-easterly wind directions and a second maximum can be found for south-westerly winds (Fig. 5c). The same wind sectors also show maximum humidity trends, though trends are larger for south-westerly than for north-easterly winds (Fig. 5g). The situation is similar in the GRL ATM box with maximum trends for both northerly and southerly winds.

In the following, we analyze trend differences for different wind sectors in more detail. For this purpose, we define off-ice and on-ice flow directions based on the wind roses presented in the previous section. In the WNB ATM box, all air masses flowing from west, north or north-west originate from sea ice covered regions. However, we only select a sector of 45° containing those wind directions for which atmospheric trends are the largest. We thus define off-ice flow as winds from 30° to 75° (north-easterly winds) for WNB and from -15° to 30° (northerly winds) for GRL. The obvious choice to define on-ice winds is then the sector directly opposite of the off-ice sectors. However, since the resulting sectors only contain a small fraction of less than 10 % of all wind directions, we decided to widen the on-ice sector to 90°. On-ice flow is then defined as winds from 165°

to 255° (south to south-westerly winds) for WNB and from 120° to 210° (south to south-easterly winds) for GRL. Sketches illustrating the off- and on-ice flow directions are presented in Fig. 5d for WNB and Fig. 6d for GRL.

Only the average wind directions in the respective ATM boxes for both regions are considered to define off- and on-ice flow conditions. We want to stress, however, that this is a simplification and that further away from the ATM box wind directions might be different. For example, when the center of a low pressure system is located over Svalbard, we would expect southerly

winds in the region east of Svalbard during conditions which were labeled as "off-ice flow" for the whole region. Thus, results from regions close to the ATM boxes are most reliable and especially the results from the region east of Svalbard should be considered with care.

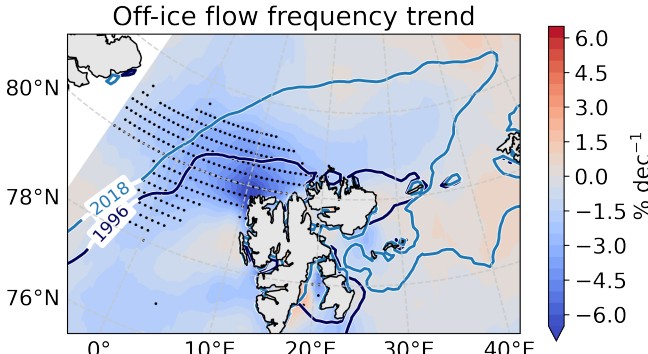

**Figure 7.** Trend map in the greater WNB region for the frequency of occurrence of off-ice flow (30°-75°) based on ERA5 wind directions for January to March of the years 1992 to 2022. Dotted areas are significant at the 95 %-level. Blue lines denote the 80 % SSM/I-ASI sea ice concentration contours averaged from January to March for two years with large and small SIE, respectively.

Off-ice flow conditions are present for about 21 % of the time in the WNB region and for about 40 % in the GRL region, which is much larger than what would be expected for a uniform distribution of wind directions (12.5 %). Trends of the

frequencies of occurrence of off-ice flow are only significant in a region north-west of Svalbard (Fig. 7) where we observe an overall reduction of up to 15 % over the 31-year period. Trends in the GRL region are not significant at the 95 %-level (not shown).

As a next step, we compare atmospheric trends during times with off- and on-ice flow and analyze in which regions they differ from each other. Figure 8 shows trend maps in the region around Svalbard. Since results based on ERA5 and MERRA-2

are very similar, we only present the ERA5 maps here. For wind speed, the differences of trends for off- and on-ice flow are very small and we thus only show maps for temperature and specific humidity. The result for wind trends in the ATM boxes are discussed later in this section.

The largest temperature trends for off-ice flow are observed directly north and north-east of Svalbard - so over the Whaler's Bay polynya region - and exceed 4 K per decade (Fig. 8a). Along the off-ice flow direction towards south-west, trends gradually

decrease but are still significant 300 km further downstream. It is interesting to note that positive air temperature trends are not restricted to open ocean regions but also extend westwards over regions with closed sea ice cover. Trends for on-ice flow are

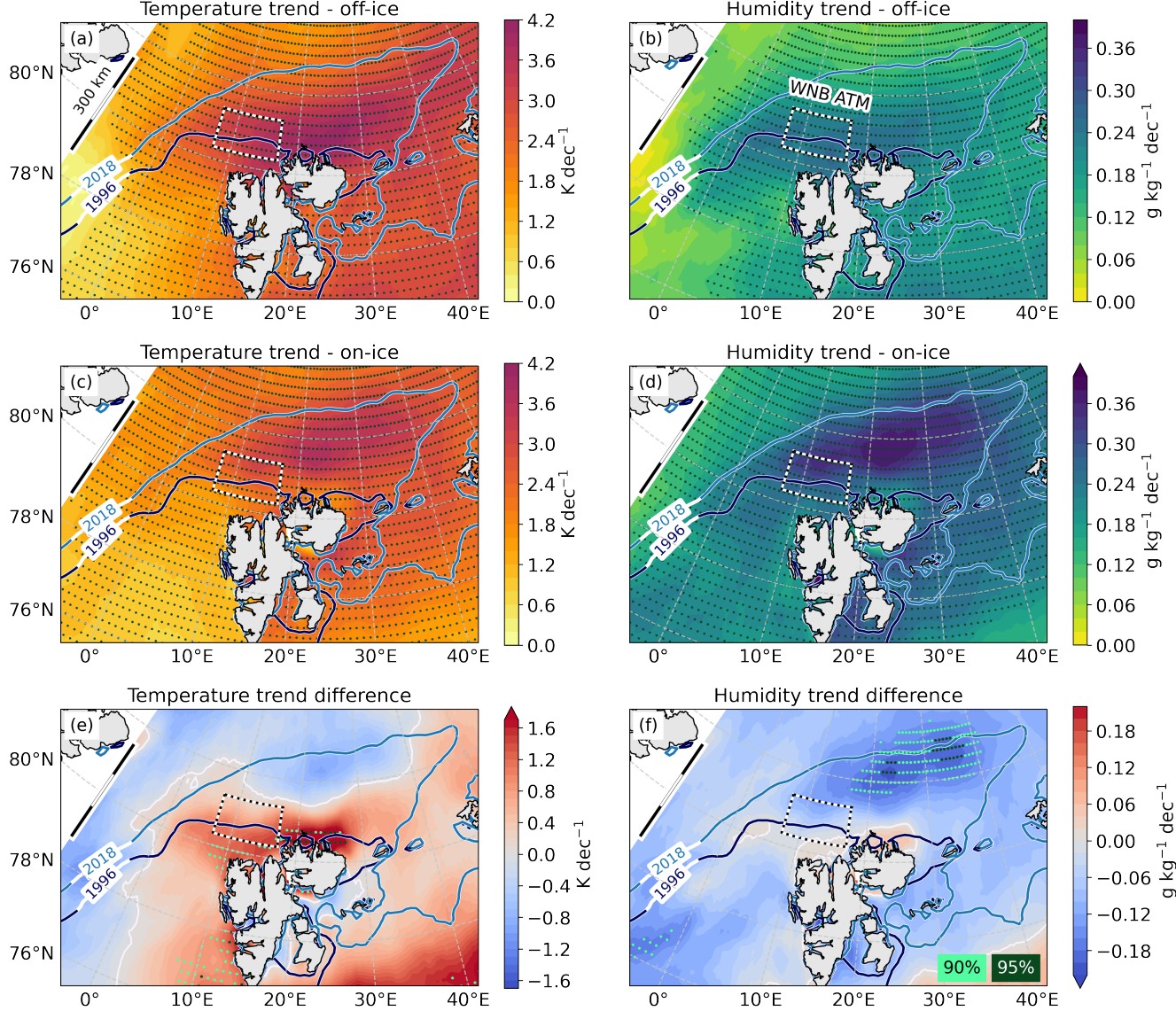

**Figure 8.** Trend maps for ERA5 air temperature (a,c) and specific humidity (b,d) for January to March of the years 1992 to 2022 using only periods with off-ice (a,b) and on-ice (c,d) winds in the WNB region. Blue lines denote the 80 % SSM/I-ASI sea ice concentration contours averaged from January to March for two years with large and small SIE, respectively. The boxes mark the location of the WNB ATM box used for time series analyses in Fig. 10 and Tab. 2. Panel (e) shows the differences of air temperature trends for off- and on-ice winds (panel (a) minus panel (c)) and panel (f) shows the corresponding trend differences for specific humidity. Dotted areas denote trends and differences of trends that are significant at the 95 %-level (dark green). For trend differences, light green dots additionally mark the 90 % significance level.

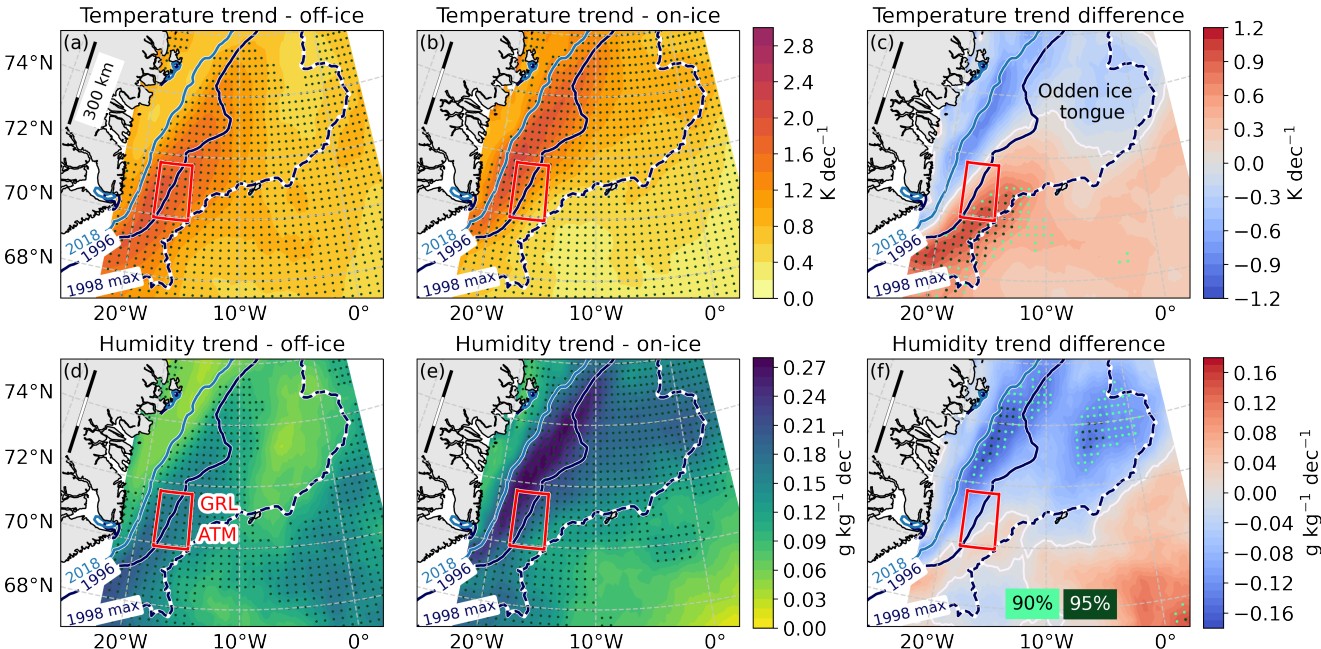

**Figure 9.** Trend maps for ERA5 air temperature (a,b) and specific humidity (d,e) for January to March of the years 1992 to 2022 using only periods with off-ice (a,d) and on-ice winds (b,e) in the GRL region. Blue lines denote the 80 % SSM/I-ASI sea ice concentration contours averaged from January to March for two years with large and small SIE, respectively. The boxes mark the location of the GRL ATM box used for time series analyses in Fig. 10 and Tab. 2. Panel (c) shows the differences of air temperature trends for off- and on-ice winds (panel (a) minus panel (b)) and panel (f) shows the corresponding trend differences for specific humidity. Dotted areas denote trends and differences of trends that are significant at the 95 %-level (dark green). For trend differences, light green dots additionally mark the 90 % significance level.

up to 2 K larger than for off-ice flow directly at the northern and western coast of Svalbard (Fig. 8e). Only these differences close to the coast of Svalbard are significant at the 90 to 95 %-level. Further towards north and west - in regions that were covered by sea ice in most of the considered years - trends are slightly larger for on-ice flow but differences to off-ice flow are
not significant.

For specific humidity, the patterns of the trends for off-ice flow look very similar to the temperature trends (Fig. 8b) with a maximum exceeding $0.25 \, \mathrm{g \, kg^{-1}}$ per decade. Humidity trends for on-ice flow are even larger and the maximum is located slightly further to the north (Fig. 8d). In this region, trends for on-ice flow are about twice as large as for off-ice flow (Fig. 8f) but unlike for temperature, we do not observe significant differences of humidity trends close to the northern and western coast
of Svalbard.

In the Greenland Sea region we find the largest trends for both considered flow directions in areas that are covered in only some years by sea ice (Fig. 9). For off-ice flow, the maximum trends are located along the 80 % sea ice contour from 1996, which was a year with large sea ice extent (Fig. 9a,d). For on-ice flow, the largest trends are located slightly further

towards the coast of Greenland than for off-ice flow (area between the 1996 and 2018 contours in Fig. 9). There, on-ice

temperature (humidity) trends are about $1 \, \text{K}$ ($0.17 \, \text{g kg}^{-1}$) per decade larger than for off-ice flow. The local maximum for off-ice temperature trends extends to the region south and south-east of Jan Mayen island, which is south of the area covered by the Odden ice tongue in some of the considered years. When comparing trends for off-ice and on-ice flow (Fig. 9c) we also see a pattern that is related to the Odden ice tongue, with larger trends for off-ice flow south of the Odden region. For humidity, trends in the Odden region are significantly larger for on-ice than for off-ice flow.

In the following, we study time series of atmospheric variables for both reanalyses in more detail. For this purpose, we present time series of January to March means for each year averaged over the WNB and GRL ATM boxes in Fig. 10 and list trends with corresponding 95 % confidence intervals in Tab. 2. In the WNB ATM box, air temperatures and specific humidity show significant positive trends with slightly smaller numbers for MERRA-2 than for ERA5 (Fig. 10) for off-ice flow. During the 31-year period, air temperatures increased by up to $10 \, \text{K}$ and the humidity by $0.7 \, \text{g kg}^{-1}$. The wind speed change is close

to zero and not significant at the 95 %-level for both reanalyses. For on-ice flow, trends for temperature are up to 50 % smaller, while trends for humidity are up to 25 % larger than for off-ice flow.

**Table 2.** Trends of meteorological variables in the WNB and GRL ATM boxes based on ERA5 and MERRA-2 reanalysis data. Trends are calculated either using only periods with off-ice or on-ice flow. Confidence intervals are calculated based on a t-test with 95 % confidence level. Bold face denotes trends that differ significantly from each other at the 90 %-level.

| Meteorological variable | ERA5 off-ice | ERA5 on-ice | MERRA-2 off-ice | MERRA-2 on-ice |
|---|---|---|---|---|
| **WNB** | | | | |
| Temperature trend ($\text{K dec}^{-1}$) | $3.3 \pm 1.4$ | $2.6 \pm 1.2$ | $\mathbf{2.9 \pm 1.5}$ | $\mathbf{1.5 \pm 0.9}$ |
| Specific humidity trend ($\text{g kg}^{-1} \, \text{dec}^{-1}$) | $0.24 \pm 0.11$ | $0.30 \pm 0.14$ | $0.23 \pm 0.13$ | $0.21 \pm 0.13$ |
| Wind speed trend ($\text{m s}^{-1} \, \text{dec}^{-1}$) | $\mathbf{0.0 \pm 0.3}$ | $\mathbf{0.5 \pm 0.4}$ | $-0.2 \pm 0.3$ | $0.0 \pm 0.5$ |
| **GRL** | | | | |
| Temperature trend ($\text{K dec}^{-1}$) | $1.7 \pm 0.8$ | $1.2 \pm 0.5$ | $1.6 \pm 0.8$ | $1.0 \pm 0.4$ |
| Specific humidity trend ($\text{g kg}^{-1} \, \text{dec}^{-1}$) | $0.16 \pm 0.09$ | $0.19 \pm 0.10$ | $0.16 \pm 0.10$ | $0.18 \pm 0.10$ |
| Wind speed trend ($\text{m s}^{-1} \, \text{dec}^{-1}$) | $-0.3 \pm 0.4$ | $0.0 \pm 0.3$ | $-0.5 \pm 0.4$ | $-0.1 \pm 0.4$ |

Temperature and humidity trends in the GRL ATM box are generally smaller than for WNB but still significant at the 95 %-level (Tab. 2). Both reanalyses agree well on the magnitude of the trends for off-ice flow with overall temperature and humidity increases of up to almost $5 \, \text{K}$ and $0.5 \, \text{g kg}^{-1}$ within 31 years, respectively. Wind speed trends are negative but only barely

significant for MERRA-2 with an overall decrease of up to $1.5 \, \text{m s}^{-1}$ within 31 years. Like for WNB, temperature trends for on-ice flow are smaller and humidity trends are larger than for off-ice flow.

Our comparison of trends calculated based on ERA5 and MERRA-2 data indicate that the results are robust with respect to the choice of reanalysis dataset. Especially in the GRL region, trend values are almost identical. The largest trend differences occur for temperature in the WNB ATM box. It is evident from Fig. 10a that MERRA-2 temperatures are up to $4 \, \text{K}$ higher than

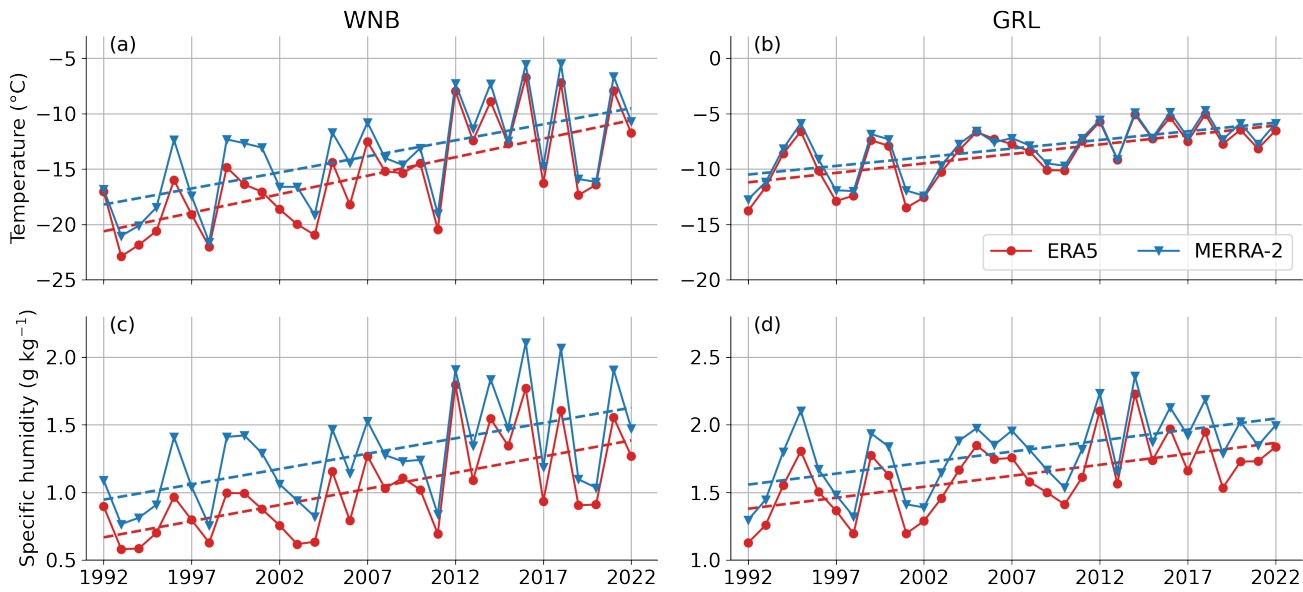

**Figure 10.** January to March averages of air temperature (a,b) and specific humidity (c,d) based on ERA5 (red) and MERRA-2 (blue) reanalyses for off-ice flow in the WNB (a,c) and GRL (b,d) ATM boxes. Dashed lines indicate trends.

ERA5 temperatures in the first half of the considered period, while differences decrease to about 1 to 1.5 K in the second half. Thus, even though trends are smaller for MERRA-2, overall temperatures in the last decade were even larger than for ERA5. A few previous studies focusing on the analysis of systematic differences between reanalyses in this region have provided similar results. Graham et al. (2019) compared reanalysis data to meteorological observations during the N-ICE campaign in winter 2015 and found that both ERA5 and MERRA-2 overestimate winter temperatures by more than 3 K. Contrary to

our results, however, biases are slightly larger for ERA5 than for MERRA-2. They also found that among all the reanalyses considered, MERRA-2 has the largest overestimation of the downward longwave radiation flux. Comparing reanalyses with remote sensing data also revealed that MERRA-2 overestimates the near-surface cloud fraction, which increases longwave cloud radiative effects (Yeo et al., 2022). More research is necessary to assess the underlying causes for these biases in more details.

In this section, we have identified regions in which atmospheric variables differ from each other for off- and on-ice flow conditions. These differences are, however, only statistically significant in very small region (see dotted areas in Fig. 8 and 9). This trend comparison was only descriptive and did not further analyze the underlying processes. As a next step, we thus study how these trends are related to the sea ice cover (Sect. 3.3).

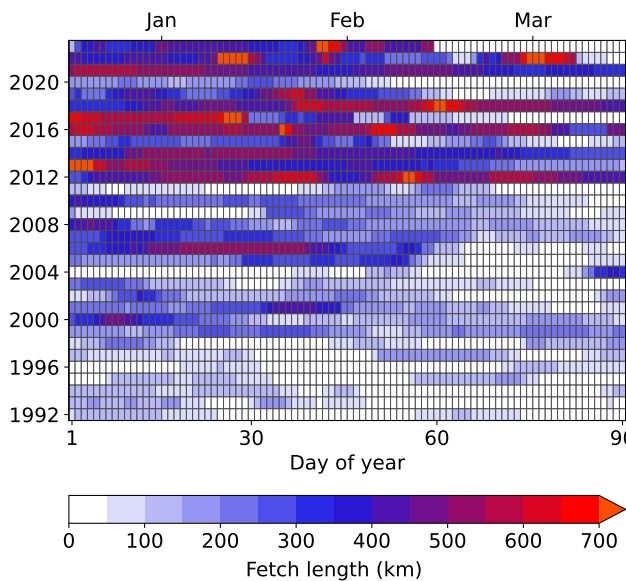

**Figure 11.** Daily fetch length over the WNB polynya region. Updated from Tetz14.

### 3.3    Assessing relationships between sea ice trends and meteorological variables

It is likely that the positive temperature and moisture trends for different flow directions analysed in Sect. 3.2 are related to different processes. During off-ice flow, cold and dry air masses from sea ice covered regions (e.g. the Central Arctic) flow towards the open ocean. These so-called cold air outbreaks (CAOs) typically occur in the Fram Strait region when a low-pressure system is present over or east of Svalbard and a high-pressure system prevails over Greenland (Knudsen et al., 2018; Kolstad, 2017). Sea ice insulates the atmosphere from the underlying ocean and thus, decreased sea ice cover causes enhanced

turbulent exchange of heat and moisture. Consequently, we expect that positive temperature and humidity trends during off-ice flow conditions show negative correlations to the upstream sea ice extent. Tetz14 found such a negative correlation for the Whaler's Bay polynya region, which weakened over the open ocean with increasing distance from the ice edge.

During on-ice flow, air masses with higher temperatures and moisture content are brought in from the south. In the Fram Strait region such warm-air intrusions are typically caused by a low-pressure system over Greenland and/or a high-pressure

system in the Svalbard region. One likely reason for the positive temperature and moisture trends found for on-ice flow in Sect. 3.2 is thus first of all a general temperature and moisture increase of the advected air-masses originating from mid-latitudes. However, southerly winds push the sea ice northward (see also Sect. C) and warmer air leads to an increased melting of sea ice. Both effects result in an increase of the open ocean area so that in turn turbulent exchange between the ocean and atmosphere is enhanced. This feedback would result in negative correlations between atmospheric and sea ice changes.

However, sensible heat fluxes are stronger during cold-air outbreaks than during on-ice flow because of larger temperature

contrasts. Despite further processes related to radiation and cloud formation (Stapf et al., 2020), we expect therefore, that during off-ice flow the local warming effect on the near-surface temperature is more effective than during on-ice flow. Especially the existence of a maximum trend in the region with reduced sea ice points to the local feedback mechanism as the dominating process.

Consequently, we analyze in this section how atmospheric changes are related to sea ice changes. We use a method inspired by the study by Tetz14, who correlated air temperatures with upstream sea ice conditions. While their analysis focused only at three locations over the open ocean with increasing distance to the ice edge, we extend their analysis not only in time, but also present maps showing correlation coefficients for each reanalysis grid point. Looking at spatial patterns instead of single points has the clear advantage that we can identify specific regions in which a substantial part of atmospheric change can be

attributed to sea ice changes.

Tetz14 calculated the polynya or fetch length as a measure for daily sea ice conditions in the WNB region until the year 2014. As described in Sect. 2.3, the WNB fetch is the distance along the green line in Fig. 1 over mostly open ocean (grid cells with a sea ice concentration below 70 %). Here, we extend their time series until the year 2022 (Fig. 11). While WNB fetch lengths exceeding 600 km did not occur in the 1990s and 2000s, such high values were much more common from 2012

onward. However, WNB fetch lengths during the last 10 years were highly variable with a more closed-up polynya in 2015 and 2020, for example. This explains the higher variability of mean sea ice concentrations in Fig. 3. Even though average sea ice concentrations in 2022 were not even among the five lowest observed years, WNB fetch values exceeded 830 km during four days in January and two days in March 2022. The event in March was a consequence of a period with strong southerly winds causing a warm-air intrusion into the Arctic across the Fram Strait region (Walbröl et al., 2023). See Fig. C1 for an example

of the flow conditions on 13 March 2022. The only other period with such high WNB fetch values was during the first days of January 2013, however, the sea ice cover quickly closed up during the next weeks in this year.

Tetz14 then analyzed ERA-Interim near-surface temperatures averaged over the winter season from January to March for the years 1992 to 2014 during off-ice flow at three grid points in the region north-west of Svalbard and found high correlations with the open-water fetch over the WNB region. In Fig. 12a we extend their analysis to the year 2022 and use a higher

temporal resolution and consider January to March monthly means instead of winter means. As an example, we show data from two points with locations close to P1 and P3 in Tetz14. We find similarly high Spearman rank correlations of $r = 0.88$ also for the extended period at a point north-west of Svalbard (see triangles in Fig. 13), which decreases to $r = 0.59$ at a point more than 200 km downstream. We performed a similar analysis to determine the relationship between average WNB sea ice concentration and atmospheric temperatures and found that correlations at the two considered points are very similar to those

using open-water fetch (Fig. 12b). Thus, instead of using the open-water fetch as in Tetz14, we consider sea ice concentrations in the ICE boxes for the following correlation analysis. By looking at single points we can only get very localized results. Not only the magnitude of correlations but also the differences between correlation coefficients for off- and on-ice flow highly depend on the considered areas (see Fig. B1 for scatter plots for off- and on-ice flow). Therefore, as a next step we calculate correlation maps between atmospheric temperatures and humidity in the region around Svalbard and the Greenland Sea and

the average sea ice concentrations in the respective ICE boxes. This allows us to identify regions where the impact of upstream

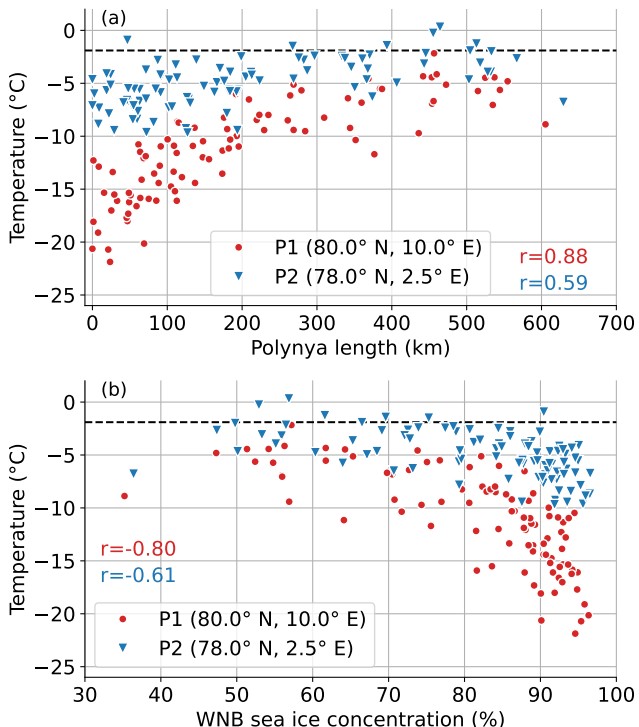

**Figure 12.** ERA5 air temperature at two locations (marked with triangles in Fig. 13) averaged monthly for January to March from 1992 to 2022 for periods with off-ice flow as function of WNB fetch (a) and of average sea ice concentrations in the WNB ICE box (b). P1 (red circles) is located close to the ice edge north-west of Svalbard and P2 (blue triangles) is located about 250 km further to the south-west over the open ocean region. Numbers are Spearman rank correlations. The dashed line indicates the freezing temperature of sea water.

sea ice concentration is largest. The squared correlation coefficient $r^2$ can also be used as a measure of the observed variance of the meteorological variables that can be explained by sea ice variability. Similar to the trends in Sect. 3.2, the maps based on ERA5 and MERRA-2 look very similar and thus we only present the ERA5 results, here.

Correlation maps for WNB for off-ice flow are presented in Fig. 13. The general patterns for both off- and on-ice flow look
very similar for temperature and humidity with the strongest negative correlations exceeding -0.85 in the area north-east of Svalbard over the center of the Whaler's Bay polynyna. Downstream of the ice edge, correlations decrease with downstream distance for both temperature and humidity until correlations are not significant any more at about 500 km downstream. For off-ice flow, we also observe smaller correlations directly at the west coast of Svalbard (at about 78.5° N, 10° E), which can be attributed to a sheltering effect of the Svalbard land mass. On the one hand, the flow likely divides around Svalbard when
winds deflect around the steep land topography. On the other hand, air masses from north-east do not experience any further warming and moistening over land and thus correlations decrease.

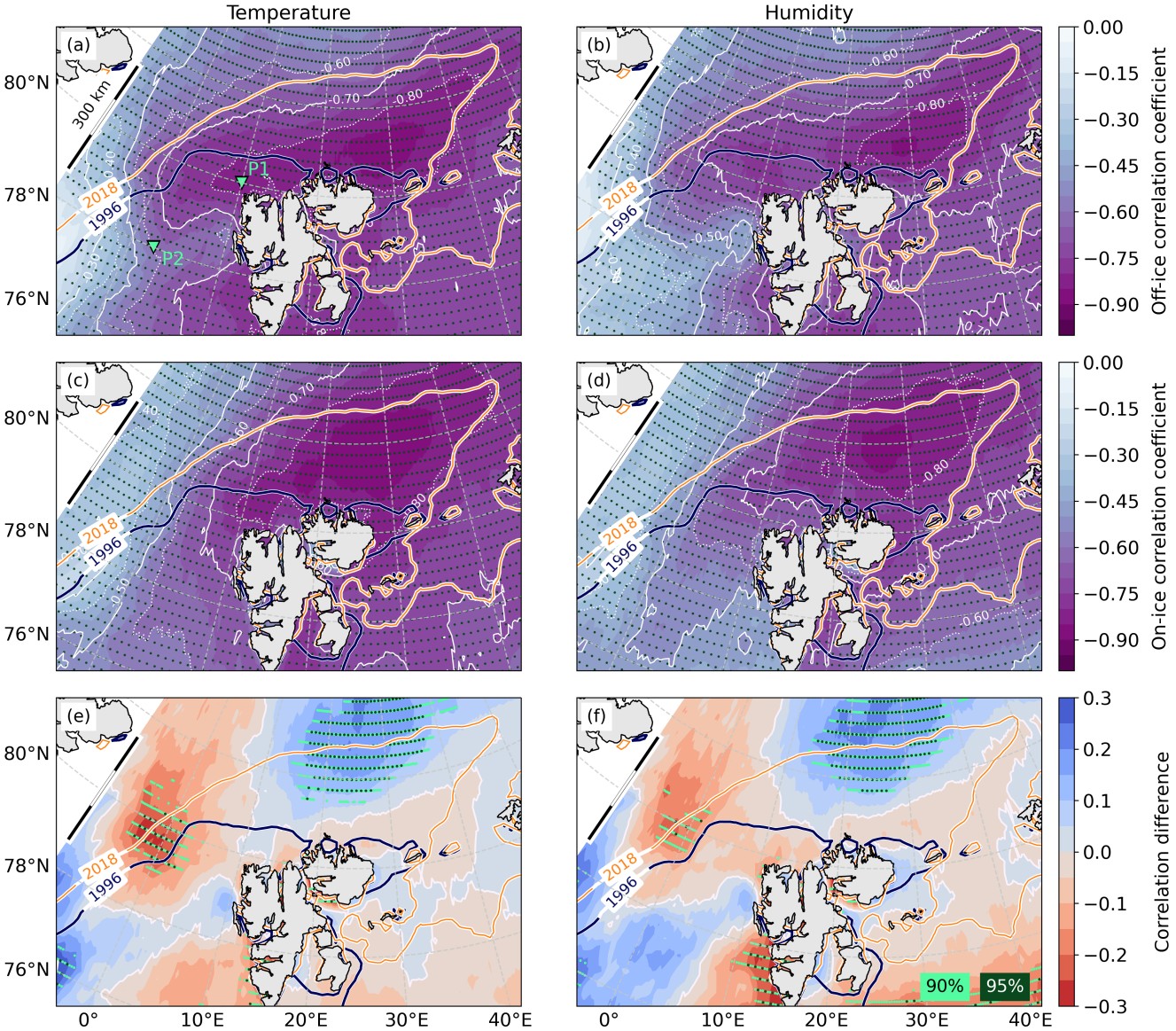

**Figure 13.** Spearman rank correlation coefficients between monthly WNB sea ice concentrations and air temperatures (a,c) and specific humidity (b,d) from ERA5 averaged for periods with off-ice (a,b) and on-ice flow (c,d). Blue and orange lines denote the 80 % SSM/I-ASI sea ice concentration contours averaged from January to March for two years with large and small SIE, respectively. Triangles indicate the locations of the points P1 and P2 used in Fig. 12. Panels (e) and (f) show the corresponding differences of correlation coefficients between off-ice and on-ice flow (panel (a) minus panel (c) and panel (b) minus panel (d)). Red colors mean that negative correlations are larger for off-ice flow compared with on-ice flow. Dotted areas denote correlations and differences of correlations that are significant at the 95 %-level (dark green). For correlation differences, light green dots additionally mark the 90 % significance level.

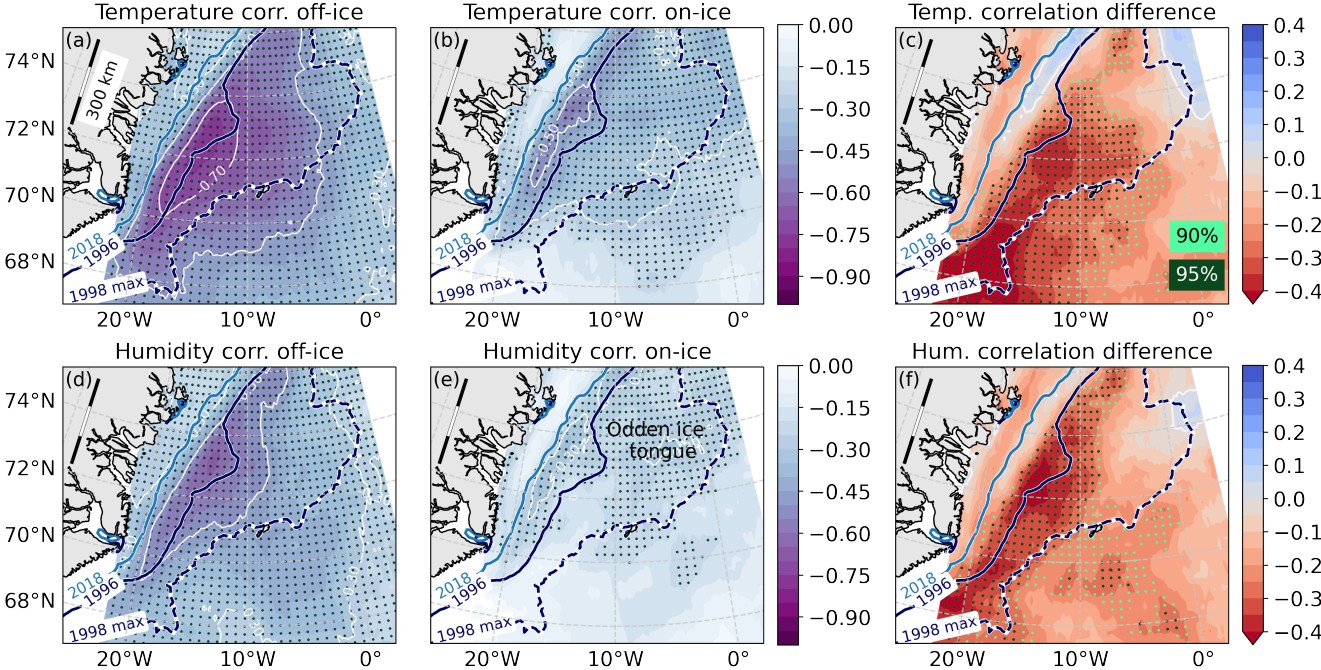

**Figure 14.** Spearman rank correlation coefficients between monthly GRL sea ice concentrations and air temperatures (a,d) and specific humidity (b,e) from ERA5 averaged for periods with off-ice (a,b) and on-ice flow (d,e). Blue lines denote the 80 % SSM/I-ASI sea ice concentration contours averaged from January to March for two years with large and small SIE, respectively. Panels (c) and (f) show the corresponding differences of correlation coefficients between off-ice and on-ice flow (panel (a) minus panel (b) and panel (d) minus panel (e)). Red colors mean that negative correlations are larger for off-ice flow compared with on-ice flow. Dotted areas denote correlations and differences of correlations that are significant at the 95 %-level (dark green). For correlation differences, light green dots additionally mark the 90 % significance level.

We also calculated the differences between the correlation maps for off- and on-ice flow for both temperature (Fig. 13e) and humidity (Fig. 13f). It is remarkable that the maps for temperature and humidity look almost identical, showing two areas of distinct differences between correlations for off- and on-ice flow. In the first one, which is located north-west of Svalbard close

to the ice edge (around 80° N, 0° E), correlations are up to 0.3 larger for off-ice flow. In this area, up to 50 % of the observed temperature and humidity variance are related to sea ice changes for off-ice flow and only about 20 % for on-ice flow. The second area is located at the northern end of the Whaler's Bay polynya (around 83° N, 25° E), where correlations are by about 0.25 larger for on-ice flow. Here the explained variance is about 40 % for off-ice flow and about two-thirds for on-ice flow.

The corresponding correlation maps for the GRL region are presented in Fig. 14. Unlike for the WNB region, correlations

are generally slightly larger for temperature than for humidity. For both flow directions, negative correlations are largest in the region that is only covered by sea ice in some of the considered years. Correlation values for off-ice flow show a maximum exceeding -0.7 for temperature and -0.5 for humidity, which is more than twice as large than for on-ice flow. This means that

up to half of the observed temperature variance close to the ice edge is related to sea ice variability for off-ice flow. Correlation differences are most pronounced and significant at the western part and south-west of the Odden region where the explained variance is up to 20 % larger for periods with off-ice than for on-ice flow (Fig. 14c,f).

Another study by Wang et al. (2019) also correlated atmospheric temperatures with sea ice conditions in all seasons and found a value of -0.46 for the whole Greenland Sea region, which is slightly smaller than our maximum values of about $r =$-0.7. Since they averaged over a much larger region and did not consider different wind directions separately, they were not able to study more local effects and specifically the impact of upstream sea ice conditions during off-ice flow.

## 3.4 Discussion of mechanisms responsible for atmospheric changes

In the previous sections we described atmospheric trends in the Fram Strait region, both in general and by looking specifically at two typical flow situations, which are cold-air outbreaks with off-ice flow and warm-air intrusions with on-ice flow. While temperature trends for both flow directions have similar maximum values but at slightly shifted locations, humidity trends are mostly larger for on-ice flow. Even though trend differences are mostly not significant we can draw valuable qualitative conclusions from the results. Firstly, most of the warming trends during on-ice flow can be explained by a temperature increase in the last decades of the air masses advected form south. This trend is comparable with the local warming due to reduced sea ice cover during off-ice flow. The increase of moisture during on-ice flow is even larger than the moistening effect over areas with reduced sea ice cover in off-ice wind conditions but also the latter is important for the observed changes.

To further analyze the impact of changes in sea ice cover on atmospheric conditions we calculated correlation maps between temperature and humidity and the respective upstream sea ice conditions for both regions. Off-ice flow consists of relatively cold and dry air masses becoming warmer and more humid the longer they are in contact with open ocean. A smaller sea ice concentration means more open water areas and thus correlations between sea ice cover and air temperature and humidity are negative. Such negative correlations for off-ice flow are found in those regions where sea ice changes are largest. However, significant correlations extend further along the off-ice flow directions. This leads to an extended area of positive correlations south of the ice edge in the GRL region and towards the south-west of the Whaler's Bay polynya. While the impact of sea ice decrease was most pronounced during off-ice flow, smaller but significant negative correlations were also found during on-ice flow. However, during on-ice flow atmospheric changes lead to sea ice changes and thus the causality is reversed compared to off-ice flow.

Our results suggest that the only region where this mechanism is significantly more important than the sea-ice related changes during off-ice flow is located at the northern edge of the Whaler's Bay polynya.

Our results hint to an important role of sea ice conditions for atmospheric warming and moistening in certain areas. However, with our present analysis we cannot exclude further possible explanations for the observed atmospheric trends. For example, cases during which warm air is advected first to the north and is flowing back to the south afterwards might impact on our correlations. We mentioned already that large scale effects (especially during on-ice flow) influence the correlations. Besides the effect due to warmer air masses from the south, for example, Wickström et al. (2020) found an increase of the winter cyclone occurrence around Svalbard from 1979 to 2016. The presence of cyclones in this region was accompanied by positive

temperature anomalies east of Svalbard. Such large scale synoptic changes certainly influence the general trends presented in Sect. 3.1, but not the analyses in Sect. 3.2 and 3.3, where results are separated by wind sectors. Warming due to large scale advection would result in much smoother and more homogeneous trend maps for the different wind direction sectors than we showed here in several figures. For example, it is evident from Fig. 8c and d for on-ice flow that trends are even larger north of Svalbard than further south, which would not be the case if trends were purely caused by trends within the inflow air masses or by altered cyclone tracks.

Our analysis also does not consider changes in sensible and latent heat transport over the ocean due to modified water temperatures that cannot be explained by sea ice decrease. The underlying processes governing the variability of ocean heat flux are very complex and not yet accurately quantified (Carmack et al., 2015; Peterson et al., 2017). Nilsen et al. (2021) found that while long-term trends in the heat transport into the Arctic Ocean could be dominated by variations in the temperature of the Atlantic Water inflow, changes on shorter time-scales are more likely due to air-ocean interaction dynamics. Their analysis indicates that wind stress is an important factor influencing the variability of the West Spitsbergen Current branches flowing over the Yermak Plateau and thus the overall heat exchange in this region.

Since the focus of this study is mainly on the impact of sea ice changes on the atmosphere, we do not disentangle whether those sea ice changes were caused by changing wind directions, increased melting by a warmer ocean or by other factors. The sea ice cover in our two study regions has changed drastically over the last 30 years resulting in a much larger area where the atmosphere comes into direct contact with the ocean without an insulating sea ice layer. Here, we assume that an increase of the open ocean area will have a much larger impact on air temperature and humidity than an increase of the ocean temperature of a few degrees in these specific regions and thus we do not consider the latter effect explicitly. We suggest that the investigation of a combined impact of ocean and sea ice conditions should be the topic of future extended work.

## 4    Summary and conclusions

The objective of this study was to examine recent regional trends in near-surface atmospheric variables in the Fram Strait region in winter and their connection to regional sea ice extent. Through the analysis of trend maps spanning from 1992 to 2022, we identified two specific focal regions where the decline in sea ice cover and the increase of atmospheric temperatures and humidity were most pronounced. The first area is located north-east of Svalbard in the Western Nansen Basin (WNB), while the second area is located in the Greenland Sea (GRL).

To assess the sea ice conditions, we calculated the average sea ice concentration from SSM/I-ASI data for the period of January to March. Our calculations focused on two designated regions, namely WNB and GRL. The WNB region exhibited a substantial decreasing trend of -10.0% per decade, accompanied by an increasing inter-annual variability within the past 10 years. Over the years, the ice edge, which was predominantly positioned at the northwest corner of Svalbard during the 1990s, has receded by more than 500 km towards the northeast during multiple days in January to March since 2012. The year 2022 stands out as an extreme case, with open water spanning over 800 km during two separate weeks. In the GRL region, the negative sea ice trend is also significant but relatively smaller at -4.7% per decade compared to WNB. This discrepancy can be

partially attributed to a higher proportion of open ocean within the GRL area already at the beginning of the analyzed period. The first half of the examined period shows the greatest inter-annual variability, likely influenced by the presence of the Odden ice tongue in certain years before 2006.

In addition, we evaluated trends in atmospheric variables across the Fram Strait region using ERA5 and MERRA-2 reanalysis data. The two datasets displayed a high level of agreement concerning the location of regional maxima and the magnitude of trends. The trend maps for air temperature and specific humidity showed very similar patterns, with the maxima roughly aligning with the areas experiencing the most substantial sea ice reductions. In the Whaler's Bay region, we observed trends of up to $3.7\,\text{K}$ and $0.29\,\text{g}\,\text{kg}^{-1}$ per decade for temperature and humidity, respectively. In the southern Greenland Sea region, the corresponding trends were up to $1.3\,\text{K}$ and $0.15\,\text{g}\,\text{kg}^{-1}$ per decade. Comparatively, trends in wind speed were generally smaller and less statistically significant than those in temperature and humidity.

As a next step, we conducted trend analyses separately for two common flow situations in the region separately and compared the outcomes. By examining wind roses, we identified wind direction sectors for off-ice flow (occurring during cold-air outbreaks) and for on-ice flow (warm-air intrusions) in both the WNB and GRL regions. While the general patterns for off- and on-ice flow exhibited some similarities, the differences between temperature and humidity trends were more pronounced in both regions. In both regions, the maximum temperature trends reached similar magnitudes, with 3-4 K per decade for WNB and 2 K per decade for GRL. However, the precise locations of the trend maxima differed depending on the prevailing wind regimes. In the WNB region, the temperature maxima consistently occurred over the Whaler's Bay polynya. For off-ice flow, they were situated near the northern coast of Svalbard and approximately 100 km further north for on-ice flow. In the GRL region, the maxima were located in areas that were covered by sea ice only in certain years, and they were approximately $100\,\text{km}$ further north for on-ice than for off-ice flow. Consequently, temperature trends were larger for off-ice flow in specific areas and larger for on-ice flow in others. The humidity trend maxima generally coincided with the temperature maxima; however, humidity trends were mostly larger for on-ice flow compared to off-ice flow. The maxima for on-ice flow were up to 50 % larger in the WNB region and almost twice as large in the GRL region compared to off-ice flow.

The presence of sea ice plays a crucial role in limiting the exchange of heat and moisture between the ocean and the atmosphere. Therefore, the fractional sea ice cover is an important factor influencing the strength of atmospheric convection and the related atmospheric warming during cold-air outbreaks. To further explore this relationship, we conducted a spatial analysis of trends in meteorological variables and their correlations with upstream ice conditions, comparing results for off- and on-ice flow. Generally, the correlation maps displayed great similarities to the trend maps. In the WNB region, the maximum correlations were located at the center of the Whaler's Bay polynya, where up to two-thirds of the observed variability was linked to sea ice changes. This impact extends hundreds of kilometers downstream of the sea ice edge. In the GRL region, the maximum correlations also aligned with the maximum trends near to the ice edge. Here, sea ice changes accounted for up to 50 % of the observed variability in temperature and 25 % in humidity. During off-ice flow, the explained variances over and downwind of the Odden region exceeded those for on-ice flow by up to 20 %.

In contrast to the trend maps where temperature and humidity exhibited differences, the disparities between correlation for off- and on-ice flow were quite similar for both variables. In the WNB region, we identified two areas with distinct differences.

Over the sea ice-covered regionNorth-west of Svalbard, correlations for off-ice flow were up to 0.3 higher. The situation wa reversed at the northern end of the Whaler's Bay polynya, where correlations for on-ice flow exceedws those for off-ice flow by 0.2. These results can be attributed to two different mechanisms. Large negative correlations for off-ice flow are caused by increased heat and moisture exchange during CAOs when the sea ice cover decreases. On-ice flow, characterized by warm and moist air during warm-air intrusions,pushes the sea ice further northward, resulting in an increased open water area and

negative correlations. In the GRL region, we found more areas in which the first mechanism dominates. Correlations for off-ice flow exceeded those for on-ice flow by up to 0.3, especially in the region south-east of the Odden ice tongue.

Our analyses demonstrate that the decreasing sea ice cover in the Fram Strait region has the most significant impact in the marginal sea ice zone during off-ice flow events, particularly during cold air outbreaks. Additionally, we observe a relationship between warm-air intrusions and decreasing sea ice cover further north. The presence of maximum trends for off-ice flow

at the sea ice margin, where sea ice changes are most pronounced, strongly indicates the crucial role of declining sea ice cover in shaping the observed atmospheric trends. one plausible explanation for these atmospheric changes is the intensified atmospheric convection and subsequent boundary-layer warming when cold and dry air from the Central Arctic flows over extended open ocean areas (e.g. Tetz14).

While our results highlight the important role of sea ice conditions in atmospheric warming and moistening, it is essential

to acknowledge that our present analysis does not exclude other potential explanations for the observed atmospheric trends. Long-term changes in large-scale synoptic conditions and circulation patterns could also influence atmospheric conditions, and changing ocean currents might contribute to increased sea surface temperatures in regions where the sea ice margin is retreating. Investigating these impact factors should be the focus of future extended research.

Our study, though regional in scope, provides valuable insights into the strong connection of atmospheric and sea ice con-

495 ditions. Some of the results can be transferred to other regions. Obviously, wind direction plays an important role for regional climate change near the marginal sea ice zones. This finding can most likely be generalized to all areas near the ice edge. Furthermore, our research highlights the large spatial variability of sea ice decline and its corresponding atmospheric response. The large trends near Svalbard align with previous studies that identify Svalbard as a hotspot in Arctic climate change. Our study contributes to a more comprehensive and detailed understanding of the ongoing changes in the Arctic.

Our analyses reveal an ongoing decrease in sea ice extent in the wider Fram Strait region, accompanied by a simultaneous increase in near-surface atmospheric temperature and humidity during winter. Exploring the related changes in the structure of the atmospheric boundary layer and in cloud processes in this region is an emerging topic, which will be partly addressed by the HALO-(AC)[3] campaign (Wendisch et al., 2021) conducted in spring 2022 in this region. Furthermore, we recommend conducting modeling studies in future research to strengthen the conclusions derived from this paper.

**Appendix A: Odden ice tongue**

The Odden ice tongue is a local phenomenon in the Greenland Sea region where sea ice episodically covers a large area north of Jan Mayen island. It occurred most frequently before 1990 (Wadhams and Wilkinson, 1999; Comiso et al., 2001) but could

still be observed in some of the years in our study period from 1992 to 2022. Figure A1 shows the mean and standard deviation of daily sea ice concentration data from SSM/I-ASI from January to March, which allows for a good visual detection of the extent of the Odden ice tongue.

## Appendix B: Correlation scatter plots for off- and on-ice flow

We demonstrated in Sect. 3.3 that during off-ice flow atmospheric temperatures northwest of Svalbard are negatively correlated to the upstream sea ice cover in the WNB ICE box (see Fig. 12). Here, we also present the corresponding scatter plots for on-ice flow condition. It is evident from Fig. B1 that it highly depends on the specific region whether correlations for off- and on-ice flow differ from each other. For a point located at the north-west corner of Svalbard (Fig. B1b) correlation coefficients are almost identical, while off-ice correlations are larger than those for on-ice flow in a region slightly further toward the west (Fig. B1a). More details about these pattern can also be seen in Fig. 13e.

## Appendix C: Synoptic conditions in 2022

2022 was a year with two periods with very low sea ice concentration in the Whaler's Bay polynya - one at the end of January and one in the middle of March (see Fig. 11). During the event in March 2022 the HALO-(AC)[3] aircraft campaign took place in the Fram Strait region. A paper describing the synoptic conditions during this period is currently under review (Walbröl et al., 2023). It identifies two distinct periods of warm-air intrusions on 12./13. March and 15./16. March causing advection of warm and moist air as far north as the Central Arctic Ocean (close to the North Pole). This effect can also be seen from the maps of temperature, humidity and wind vectors from ERA5 for a day during this period presented in Fig. C1. Southerly winds push the sea ice northward and thus are one of the factors causing a large open water area north of Svalbard.

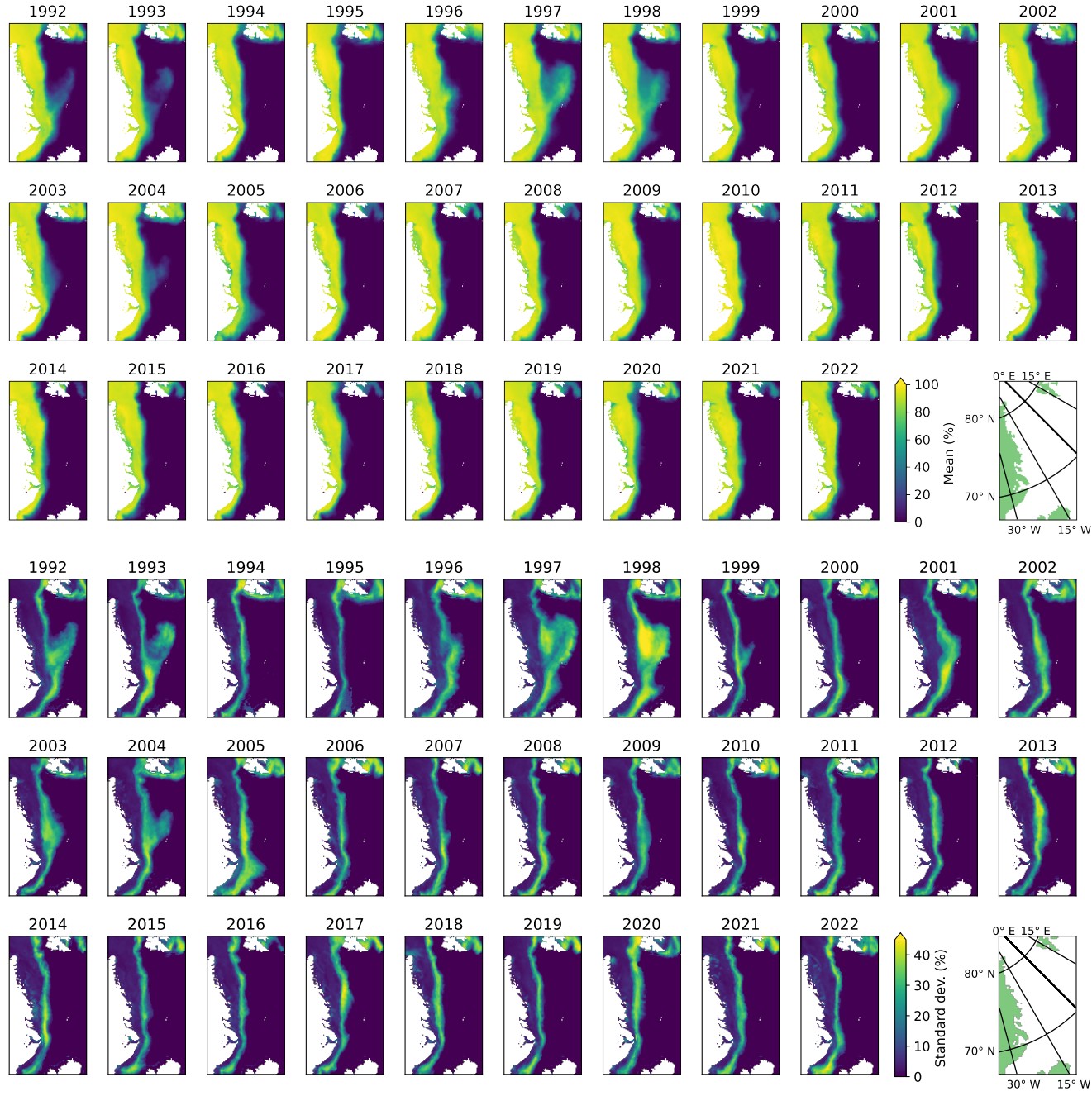

**Figure A1.** Mean (top) and standard deviation (bottom) of daily SSM/I-ASI sea ice concentrations from January to March of each year in the Greenland Sea and Fram Strait regions.

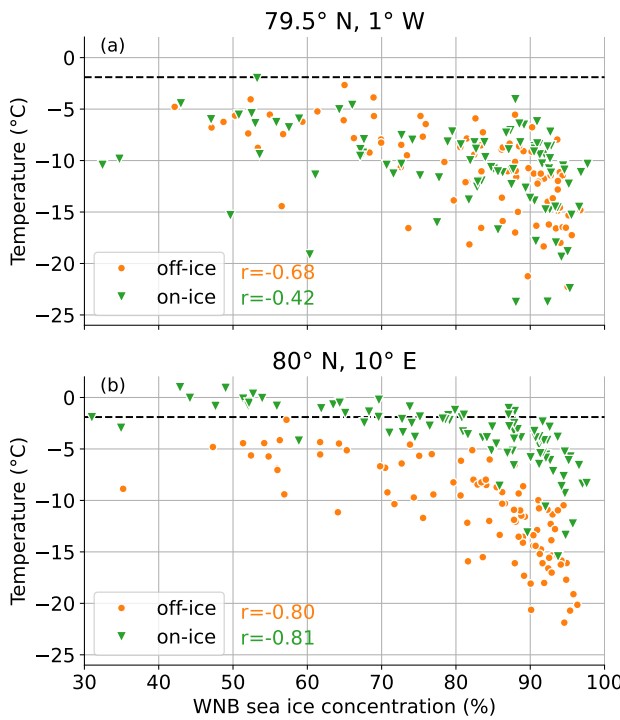

**Figure B1.** ERA5 air temperature averaged monthly for periods with off-ice (orange) and on-ice (green) flow as function of average sea ice concentrations in the WNB ICE box. Numbers are Spearman rank correlations. Results are shown for a point located in the area where correlation differences in Fig. 13e,f are largest (a) and for the more northern point indicated in Fig. 13a (b). Note that the orange dots in (b) are the same as the red dots in Fig. 12b. The dashed line indicates the freezing temperature of sea water.

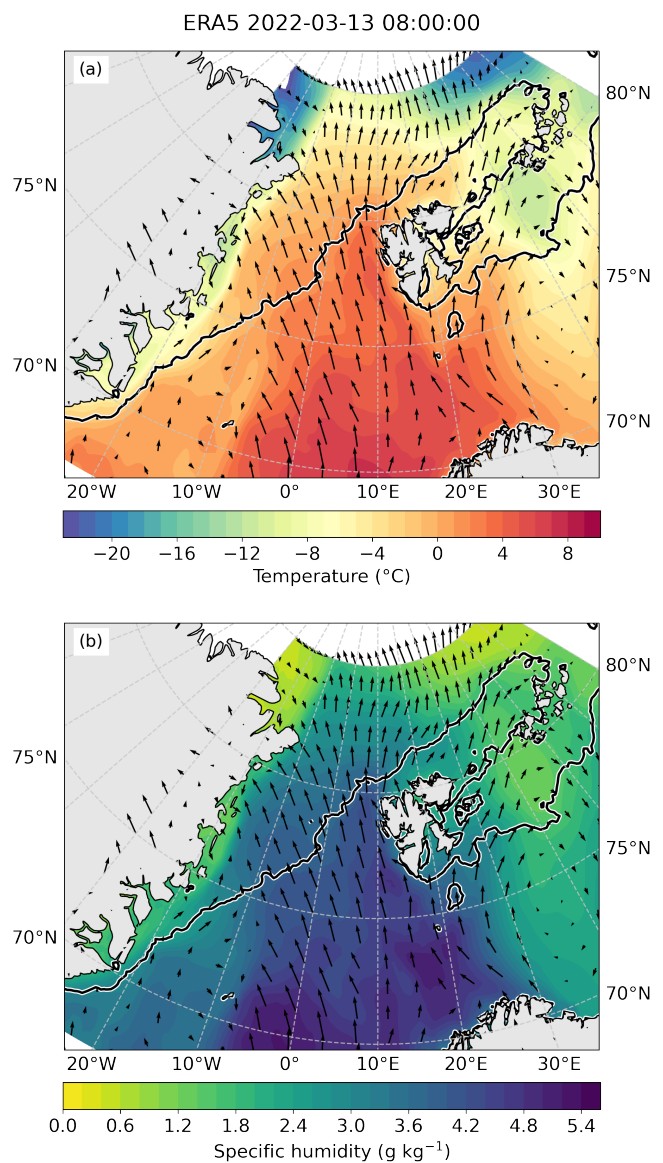

**Figure C1.** Contours of air temperature (a) and specific humidity (b) at about 10 m height from ERA5 reanalysis data on 13 March 2022. Arrows represent the corresponding wind vectors. The black line denotes the 80 %-sea ice concentration contour from the same day.

*Code and data availability.*  Kaleschke, L., F. Girard-Ardhuin, G. Spreen, A. Beitsch, and S. Kern, ASI Algorithm SSMI-SSMIS sea ice concentration data, originally computed at and provided by IFREMER, Brest, France, were obtained as 5-day median-filtered and gap-filled product for 1992–2021 from the Integrated Climate Data Center (ICDC, icdc.cen.uni-hamburg.de, University of Hamburg, Hamburg, Germany). MERRA-2 data were provided by the Goddard Earth Sciences Data and Information Services Center (https://disc.gsfc.nasa.gov/).

ERA5 profile data on model levels were obtained from ECMWF's MARS tape archive. The analysis scripts are available from the authors on request.

*Author contributions.*  AS and CL developed the original idea. AS analyzed and plotted the data. AS and CL interpreted the results and wrote the manuscript.

*Competing interests.*  We declare that no competing interests are present.

*Acknowledgements.*  This work was partly funded by the Deutsche Forschungsgemeinschaft (DFG, German Research Foundation) under Germany's Excellence Strategy – EXC 2037 'CLICCS - Climate, Climatic Change, and Society' – Project Number: 390683824, contribution to the Center for Earth System Research and Sustainability (CEN) of Universität Hamburg. It was also funded by the Deutsche Forschungsgemeinschaft (DFG; German Research Foundation) Project 268020496 TRR 172, within the Transregional Collaborative Research Center Arctic Amplification ((AC)[3]). The results contain modified Copernicus Climate Change Service information (2014-2018). Neither the Euro-

pean Commission nor ECMWF is responsible for any use that may be made of the Copernicus information or data it contains.

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
