# Peer review of "Attributing near-surface atmospheric trends in the Fram Strait region to regional sea ice conditions"

_The Cryosphere, 2022_

## Author Comment (AC1)

Dear referee,

thank you for your comprehensive review and helpful suggestions. Since your general comments contain many points dealing with similar issues, we decided to re-group some of them by specific topics instead of providing point-by-point answers to each of them. If not stated otherwise, all references to figures refer to the new figures in this document and not to the figures in the original paper draft.

1) Your first issue concerns a better explanation of the general idea of our method and the separation of flow directions. Specific suggestions were to include a schematic and to start with a more general presentation of trends by using e.g. wind roses.

We think that these are very helpful suggestions and will include a new results section (new 3.1) concerning general trends in the Fram Strait region.

- Starting with the map of sea ice concentration trends (original Fig. 6), we identify regions with the largest sea ice decline and define the specific study regions WNB and GRL and the corresponding ICE boxes used in later sections.
- Afterwards, we present trend maps (without any dependence on wind direction) for temperature, humidity and wind speed for both ERA5 and MERRA-2 data. Exemplary plots for ERA5 temperature and humidity trends are shown in Fig. 1.

[Figure]

Figure 1: Trends of air temperature (left) and specific humidity (right) based on ERA5 data from 1992 to 2022. Dotted areas are not significant at the 95 %-level. Blue lines denote the 80 % SSM/I-ASI sea ice concentration contours averaged from January to March for two years with large and small SIE, respectively.

- For both regions (WNB and GRL) we look specifically at that area where the ice edge was located in years with a large sea ice cover (e.g. 1996). We then place the ATM boxes in the location with the largest temperature and humidity trends in these areas (see Fig. 2d) and

calculate wind roses for the ATM boxes. Examples for temperature are presented in the following figures:

[Figure]

Figure 2: Wind roses for the WNB ATM box based on ERA5 data. The length of the bars indicates the frequency of occurrence within each wind direction bin. The colors indicate average temperatures (a,b) for the first and last 10 years of the study period, and temperature trends over the 31 years (c). Straight dark blue solid lines indicate the off-ice sector and blue dashed lines the on-ice sector (c,d). Sketch illustrating the placement of the ATM box and the wind direction sectors for on- and off-ice flow (d). The background shading shows ERA5 temperature trends (zoom into Fig. 1).

[Figure]

Figure 3: Same as Fig. 2 but for GRL.

- Wind sectors are now chosen according to the following criteria: For off-ice flow, we look at those wind directions for which the flow generally originates from an area covered by sea ice. From those, we select a sector of 45° for which temperature and humidity trends are largest (Fig. 2c and 3c). This results in off-ice sectors of 30°-75° (north-east) for WNB and of 345° to 30° (north) for GRL. For on-ice flow, one could simply assume the opposite sectors - i.e. south-east for WNB and south for GRL. However, these sectors contain only about 5-6 % of all cases and thus we extend the size of the on-ice sector to 90°. The new on-ice sectors then span from 165° to 255° (south-west to south) for WNB and from 120° to 210° (south to south-east) for GRL. An illustration of the sectors and flow directions is shown in Fig. 2d and 3d, which hopefully helps better illustrate the concept of off- and on-ice flow.

- The results for trends and correlations for different flow directions, which were presented in the old Sect. 3.1 and 3.3, are then reprocessed using this new definition of off- and on-ice flow. In the GRL region, this results in only small changes (original Fig. 4 and 11). In the WBN region, changes for temperature are notable but do not change the overall conclusions (Fig. 4 e). The largest changes occur for humidity trends in the WNB region. With the new definition of wind direction sectors, humidity trends are now larger for on-ice flow in all regions (Fig. 4b,e,f). However, there are hardly any changes in the correlation maps (Fig. 5), which look almost identical for temperature and humidity.

[Figure]

Figure 4: Trend maps in the greater WNB region for ERA5 air temperature (a,c) and specific humidity (b,d) for January to March of the years 1992 to 2022 using only periods with north-easterly winds (a,b) and using periods with all other wind directions (c,d). Black dotted areas are not significant at the 95 %-level. Blue lines denote the 80 % SSM/I-ASI sea ice concentration contours averaged from January to March for two years with large and small SIE, respectively. Panel (e) shows the differences of air temperature trends using north-easterly winds and the trends using all other wind directions (panel (a)

minus panel (c)) and panel (f) shows the corresponding trend differences for specific humidity. White dotted areas in e and f are significant at the 95%-level.

[Figure]

Figure 5: Differences of correlation coefficients between sea ice concentration and (a) temperature and (b) humidity trends (off-ice minus on-ice flow). White dotted areas are significant at the 95 %-level. Blue lines denote the 80 % SSM/I-ASI sea ice concentration contours averaged from January to March for two years with large and small SIE, respectively. Green triangles mark the locations used in Fig. 6.

2) The second issue concerns a generally better explanation of the methodology. What are the implications of the differences between off-ice and on-ice flow? Why is upstream sea ice cover chosen? How can the effect of sea ice cover be entangled from large-scale atmospheric warming or increased ocean heat?

Generally, we will address these issues by slightly adjusting the storyline and the overarching questions of the paper. This different framing of the topic helps to clarify the applied methods. The new structure is derived from the following questions:

1. What are the trends of near surface atmospheric variables in the Fram Strait region?

2. Two typical synoptic situations in this region are cold air outbreaks (CAOs) with cold, dry air flowing from sea ice covered regions towards the open ocean and warm air intrusions bringing in warm and moist air during southerly flows. What atmospheric trends can we observe for these two situations – off-ice flow during CAOs and on-ice flow during warm air intrusions?

3. One factor influencing the strength of CAOs in this region is the open water fetch or the fraction of ocean covered by sea ice. Can we find a relationship between the atmospheric trends over the open ocean in the Fram Strait region and a decrease of sea ice in the upstream region? We address this question by calculating correlations between sea ice cover and atmospheric variables for both on-ice and off-ice flow. During off-ice flow, a decreased sea ice cover means larger fluxes of heat and moisture towards the atmosphere. We would expect negative correlations in this case. During on-ice flow, winds influence the sea ice drift and push the ice edge further to the north/north-east. Furthermore, warm air masses can cause a melting of sea ice, also reducing the sea ice cover. This would also lead to negative correlations.

In regions where negative correlations are stronger for off-ice flow than for on-ice flow the first mechanism likely dominates and vice versa.

We will change the introduction accordingly and also include this information in short paragraphs at the beginning of each section in the results chapter.

- Large scale atmospheric processes can also have an impact on observed atmospheric trends. For example, Wickström et al. (2020) found an increase of the winter cyclone density around Svalbard from 1979 to 2016. The presence of cyclones in this region was accompanied by positive temperature anomalies east of Svalbard. Such large scale synoptic changes certainly influence the trends presented in Fig. 1 above but not the following analyses, where results are separated by wind sectors. It is also possible that the air flowing in from the Central Arctic (for off-ice flow) or from more southern latitudes (for on-ice flow) have become warmer and more humid over time. Such an effect, however, would result in much smoother and more homogeneous trend maps for the different wind direction sectors. For example, it is evident from Fig. 4c and d above for on-ice flow that trends are even larger north of Svalbard, which would not be the case if trends were purely caused by trends within the inflow airmasses.

- An increased inflow of warm water also influences the heat transfer from the ocean to the atmosphere and can thus have an impact on atmospheric trends. Increased ocean heat also causes melting of sea ice. The underlying processes governing the variability of ocean heat flux are very complex and not yet accurately quantified (Charmack et al., 2015). Nielsen et al. (2021) found that while longterm trends in the heat transport into the Arctic Ocean could be dominated by variations in the temperature of the Atlantic Water inflow, changes on shorter time-scales are more likely due to air-ocean interaction dynamics. Their analysis indicates that wind stress is an important factor influencing the variability of the West Spitsbergen Current branches flowing over the Yermak Plateau and thus the overall heat exchange in this region. Thus, it is difficult to look at an isolated ocean impact separately. However, in this study we focus mainly on the impact of sea ice changes on the atmosphere. We do not disentangle whether those sea ice changes were caused by changing wind directions, increased ocean melting or by other factors. Since sea ice insulates the ocean from the atmosphere and greatly reduces heat and moisture exchange, we assume that an increase of the open ocean area will have a much larger impact on air temperature and humidity than an increase of the ocean temperature of a few degrees.

We will now include a detailed discussion of these impacts.

**General comments**

As I understand it, the main objective of the study is to distinguish impacts on

atmospheric temperature and humidity due to "regional sea ice changes" vs "other factors influencing atmospheric conditions". The separation between the two is done by looking separately at periods of "on-ice" and "off-ice" winds, defined by specific wind direction ranges. I was occasionally confused by this approach; it seems to me that much more space needs to be given to explaining the idea behind this method and what the implications of the observed differences actually are.

See 2).

From what I understand, the authors look to separate temperature/humidity trends due to "effects of changing sea ice cover" vs "other factors" by which they seem to imply warm intrusions from the south. It is not obvious to me that the "on/off-wind" separation is a good way to do this (wouldn't it be easier to single out southerly winds, for example, or to look at heat transports directly?). Decreased regional sea ice cover presumably does not require "off-ice flow" to affect air temperature/humidity near the ice edge – I am sure the authors are aware of this, but I believe that they should lay out the motivation behind their methodology much more clearly. Moreover, it is not clear to me that the "off-ice" direction necessarily corresponds well to actual off-ice flow.

We adjusted the off-ice wind sectors and contrast them now with southerly on-ice flows. See 1) for details.

The manuscript lacks a thorough discussion of confounding variables, and I often had trouble with the inferred causality. For example, take the statement of L309: ".. correlations for off-ice flow exceed -0.8 for both temperature and humidity and this S.I.C. changes in the upstream region can explain up to two thirds of the observed [temperature and humidity] variability..". Would they get a different result if they replaced the WNB box with a box to the NW, or just used the same box as for the atmosphere? If not, what are the implications for this statement? And what about large-scale atmospheric warming or increased ocean heat; wouldn't that affect both variables? I understand that it is hard to pick apart the many interwoven mechanisms at play, but I am missing a more clear and thorough discussion of exactly what the authors have found.

A clearer discussion will be included. See 2) for details.

I do not recommend the publication of this manuscript in its current form; my recommendation is that the paper undergoes major revisions, which I strongly suggest should include a comprehensive overhaul of the paper with the goal of making it much clearer to the reader why the particular approach was chosen, and what one should actually make of the results. I personally suspect that it might be beneficial to separate both the results and discussion into one section dealing with general trends/correlations and another dealing with the difference between on-ice/off-ice winds (although that would certainly not be the only way to go about it).

We now start with a presentation of general trends, followed by a comparison of cases with off-ice and on-ice flow. See 1) for details.

I recognize that the mechanism the authors invoke is somewhat complex, and that I may be missing important aspects out of ignorance. If that is the case, I hope the authors take my input as motivation for providing a clearer framing of the study in a future version of the manuscript.

**Specific comments**

- The authors need to explain more clearly how their methodology of looking at trends/correlations during different wind directions relates to main objective of the study (separating effects of regional sea ice loss vs "other effects"/southerly heat transport). Perhaps some sort of schematic could be helpful?

The method will be explained in more detail (see 2) for details). A schematic will be included (see Fig. 2d and 3).

- The study relies on the separation between "on-ice" vs "off-ice" winds. These are defined as specific wind direction ranges for the two regions. In my opinion, this choice needs to at least be justified more clearly. For example, would not winds from the NW be more directly "off-ice" at the WNB ATM box than those from the NE? And area the "WNB ICE" box and "fetch line" actually upwind of the "WNB ATM" box during off-ice winds per this definition? It is possible that all of this would be more obvious to a reader more intimately familiar with the region than I am. However, I think it might be helpful in this respect to show some context at the start of the paper; e.g., wind roses with temperature and/or humidity, distribution of temp/hum as functions of wind direction, or map plots of the mean wind/temperature fields during on/off-ice winds might help setting the stage.

Thank you for the suggestion. We think that wind roses are very helpful to present an overview of conditions during different flow directions. They will be included in the results (Fig. 2 and 3). See 1) for details – also concerning the placement of the ATM and ICE boxes.

- It should also be made explicit, or at least discussed in more detail, whether off-ice winds cause a redistribution of heat/moisture within a larger region, or whether this is a mechanism that has caused net increases in heat/moisture in the Barents/Fram Strait area. It was not clear to me from e.g. the trend/correlation difference plots (Fig 3ef, 4cf, 10ef, 11cf) whether the positive/negative regions actually balance out.

All trends for temperature and humidity are positive regardless of wind direction (see Figures 1 and 4), which opposes the hypothesis of a redistribution of heat and moisture in this region.

- I think the authors need to state more clearly whether the differences between trends during "off-ice" and "on-ice" wind conditions are actually statistically different. Does Table 2 indicate that they are not? If so, how does that impact the conclusions?

We now calculate a t-test to compare the significance of the difference of two different trends with slopes $b$ and standard errors $s_b$ using the following formula:

$$t = \frac{b_1 - b_2}{s_{b1}^2 + s_{b2}^2}$$

with $df = n_1 + n_2 - 4$ degrees of freedom. Generally, the trends in Table 2 are significantly different only at the 80%-level or not at all. We also indicated areas where trend differences are significant at the 95%-level in the corresponding maps (see Fig. 4e,f). There, also only a small fraction is significant. With the refinement of the research questions and the overall structure of the paper described in 1) and 2) above, however, it is not the main focus of this section – where we analyze trends for on-ice and off-ice flow – to detect whether trends are significantly different. The aim is rather to compare the magnitude of the trends. It is then the goal of the following section to investigate the impact of the sea ice cover on atmospheric conditions. Our analysis of correlations differences already included a significance test.

- Throughout the manuscript, there needs to be a clearer differentiation between trends/effects that are attributed specifically to "off-ice flow" vs to other effects/"general trends".
  One specific example: From L5 in the abstract ("During off-ice flow.."): It seems necessary here to include the corresponding temperature changes during the other wind directions. Another example: Red markers in Figure 9 show the relationship between air temperature NW of Svalbard as a function of WNB S.I.C./polynya length. How different would this figure look if you only included "on-ice" winds?

We will describe the differences between on-ice and off-ice flow in more detail. See also 1) and 2).

Figure 6 is similar to Fig. 9 in the original manuscript, but this time also includes off-ice flow. The left panel shows the more northern of the original two points. Generally, the shape change of temperature with sea ice concentration is very similar for both on-ice and off-ice flow and also correlation values do not differ much. It is evident from Fig. 5 that this point is located in an area where we do not observe large differences in correlations. The right panel thus shows a second point located further west, for which correlations are lower for on-ice flow. Also at this point, the temperature change with sea ice concentration looks very similar for both wind sectors. However, since correlation does not mean causation, different processes are probably related to the observed behavior.

For off-ice flow, reduced sea ice cover allows for an increased heat exchange with the ocean and thus increases air temperatures. Strong on-ice flow generally pushes the ice edge further north and thus decreases sea ice cover. In addition, warmer air increases melting of sea ice. We will discuss these mechanisms in more detail in the corresponding section of the paper.

[Figure]

Figure 6: ERA5 air temperature at two locations averaged monthly for periods with off-ice and on-ice north-easterly flow as function of WNB sea ice concentration. Numbers are Spearman rank correlations.

- L129: The formula by Steiger 1980, or a brief description of what it is, should be included.

The formula is based on Fisher's z-transform, where both correlation coefficients *r* are transformed to a *z*-value:

$$z = \frac{1}{2}\ln\left(\frac{1+r}{1-r}\right)$$

and a t-test is applied to the difference $z_1 - z_2$. We will include this description in the paper.

- Figure A1: I am a little confused as to why the SD of sea ice concentration is shown here. Why not just show the actual (winter average) concentration?

In our opinion, the outline of the Odden ice tongue can be visually detected more clearly from the standard deviations than from the mean sea ice concentration. This can be seen from Fig. 7. Nevertheless, we will include maps of the means as an additional figure in the appendix.

[Figure]

Figure 7: Mean (top) and standard deviation (bottom) of January to March SSM/I-ASI sea ice concentrations.

- Great that the study looks at two different reanalysis products, this strengthens the analysis. Figure 5 shows apparent striking systematic differences in both temperature and humidity – it would be useful if the authors could briefly comment on possible reasons for this (different height levels? known biases?).

We use MERRA-2 data from provided at 10 m height and ERA5 from the lowest model level, which varies roughly from 8 to 10 m height in this region and season. Such a small height difference between the two reanalyses does not explain the observed differences. A few studies have focused on the analysis of systematic differences between reanalyses in this region. Peterson et al. (2017) found that ERA5 has a stronger overestimation of winter temperatures, while MERRA-2 has a stronger overestimation of the downward longwave radiation flux. Yeo et al. (2022) showed that MERRA-2 exhibits a stronger overestimation of near-surface cloud fraction, which increased longwave cloud radiative effects. Overall, more

research is required to identify the causes of these reanalysis biases. Nevertheless, we will briefly comment on this issue in the corresponding section of the paper.

In general, I found the figures to be nicely made and helpful. I would suggest a few modifications:

- Add scale bars to the maps (helps to interpret statements like "500 km downstream", etc).
- Clearly label the boxes – e.g. GRB (ICE), GRB (ATM) or similar; it is at times difficult to follow which is which.
- Label the Odden Ice Tongue somewhere in Figure 1. The Odden ice tongue should also be indicated in Fig. 4.
- Revise the colormaps such that warm colors correspond to warm temperatures etc (this would make especially Figures 3/4 a bit more intuitive). Figures 3/10 and 4/11 should at least have the same color showing the same sign of temp/hum change.

The maps now include scale bars (see Fig. 1 and 4) and all boxes will be labeled more clearly (see Fig. 4). The Odden ice tongue (and also the Yermak plateau – see next point) will be labeled in the respective figures.

We now omit the purple end of the colormap for temperature trends so that red colors correspond to regions with the largest warming. The original Fig. 10 and 11 do not show trends but correlation coefficients. We originally chose to use the same color maps as in the original Fig. 3 and 4 to make it easy to distinguish results for temperature and humidity. However, we see that this might be misleading and will now use a different colormap for all correlation plots (original Fig. 10 and 11 a-d) ranging from purple for strong negative correlations to light blue for a correlation of 0.

- The area NW of Svalbard where trends are most affected by off-ice winds (e.g. Fig. 3ab, Fig3ef) seems to correspond roughly to the Yermak Plateau, which from what I understand is an area where the upper ocean is particularly warm and loses a lot of heat and moisture to the atmosphere. Could this play a role in the mechanism that the authors invoke? (Note: I don't expect the authors to go into detail, but I think it warrants a mention).

With the updated definition of wind direction sectors the original Fig. 3 has changed a bit in this specific region (see Fig. 4) so that positive trends do not extend as far to the north-west as before. Nevertheless, we checked whether an impact of the Yermak plateau is notable in the correlation maps (see Fig. 8). It is evident that the area of maximum correlation differences for on-ice and off-ice flow do not align very well with the outline of the Yermak plateau and thus it is unlikely that local heat exchange due to ocean mixing dominates the observed trends for off-ice flow.

Nevertheless, we will mention the general importance of the Yermak plateau on the ocean circulation in this region. See also 2) for details.

[Figure]

Figure 8: Same as Fig. 5a but with an added isoline at 1500m depth to illustrate the location of the Yermak plateau. Based on GEBCO bathymetry data.

**Technical/minor**

- L20-22: Meaning is unclear.

The sentence will be rephrased to:

"The strength of the trend also highly depends on the region. Stroeve and Notz (2018) and Onarheim et al. (2018) found that the regions with the largest decrease of SIE were the Beaufort and East Siberian Seas in September and in the Barents and Greenland seas in March."

- L147 and onward: "trends of the frequencies" – I find this use of "frequency" confusing (others may not)

What we mean here is the "frequency of occurrence". This term will be used for clarity.

- L142. "Westerly to northerly": If this refers to the 30-60 degree window, this phrasing seems inaccurate.

This sentence refers to the -45° to 15° window.

- L173: "time series of trends" – meaning unclear

It should read "time series of atmospheric variables" and will be corrected.

- I would advise being careful with the use of "as for"; to me, this reads as "with regard to". (Ex. L189, L245, L280).

We will rephrase to

L189: "Like for WNB, trends are calculated using ..."

L245: "Similar to the atmospheric trends (Sect. 3.1), the general patterns ..."

L280: "This is almost one third larger than trends calculated using periods with all other wind directions, which is a similar result as for WNB."

**References**

Carmack, E., Polyakov, I., Padman, L., Fer, I., Hunke, E., Hutchings, J.,et al.  (2015). Toward quantifying the increasing role of oceanic heat in sea ice loss in the new Arctic. Bulletin of the American Meteorological Society, 96(12), 2079-2105, 10.1175/BAMS-D-13-00177.1

Graham, R. M., Cohen, L., Ritzhaupt, N., Segger, B., Graversen, R. G., Rinke, A., Walden, V. P., Granskog, M. A., and Hudson, S. R. (2019): Evaluation of Six Atmospheric Reanalyses over Arctic Sea Ice from Winter to Early Summer, *Journal of Climate*, *32*(14), 4121-4143, 10.1175/JCLI-D-18-0643.1

Nilsen, F., Ersdal, E. A., & Skogseth, R. (2021). Wind-driven variability in the Spitsbergen Polar Current and the Svalbard Branch across the Yermak Plateau. Journal of Geophysical Research: Oceans, 126, e2020JC016734, 10.1029/2020JC016734

Peterson, A. K., Fer, I., McPhee, M. G., & Randelhoff, A. (2017). Turbulent heat and momentum fluxes in the upper ocean under Arctic sea ice. Journal of Geophysical Research: Oceans, 122(2), 1439-1456, 10.1002/2016JC01228

Wickström, S., Jonassen, M. O., Vihma, T., & Uotila, P. (2020). Trends in cyclones in the high-latitude North Atlantic during 1979–2016. Quarterly Journal of the Royal Meteorological Society, 146(727), 762-779, 10.1002/qj.3707

Yeo, H., Kim, M. H., Son, S. W., Jeong, J. H., Yoon, J. H., Kim, B. M., & Kim, S. W. (2022). Arctic cloud properties and associated radiative effects in the three newer reanalysis datasets (ERA5, MERRA-2, JRA-55): Discrepancies and possible causes. Atmospheric Research, 270, 106080, 10.1016/j.atmosres.2022.106080

---

## Author Comment (AC2)

Dear referee,

Thank you for your helpful comments. Please find point-by-point answers below.

1) How could the effects of ocean can be separated from that due to pure ice changes?

The ocean can impact near atmospheric conditions in two ways. Directly – in areas with open ocean – by increasing heat and moisture fluxes to the atmosphere. And indirectly by melting sea ice, which acts as a barrier between ocean and atmosphere and largely reduces heat and moisture exchange. Since we do not study the processes causing changes in sea ice cover, it is not possible to exclude the indirect ocean impact as a factor influencing atmospheric conditions. The sea ice cover in our two study regions has changed drastically over the last 30 years resulting in a much larger area where the atmosphere comes into direct contact with the ocean without an insulating sea ice layer. Here, we assume that an increase of the open ocean area will have a much larger impact on air temperature and humidity than an increase of the ocean temperature of a few degrees in these specific regions and thus we do not consider the direct ocean effect explicitly.

2) Would you mind considering using a model to confirm the main conclusions of this study?

We agree, modeling would strengthen the conclusions, but this is beyond the scope and possibilities of this paper. It would require a very comprehensive analysis. In fact, one of the goals of this paper is to stimulate future modeling work based on our present results. We will add this idea to the outlook paragraph at the end of the section *Discussion and conclusions.*

3) Do the off-ice events have associations with typical large-scale atmospheric circulations? Or is there any connection with synoptic cyclones?

Events of off-ice flow in the Fram Strait region typically occur when a low-pressure system is present over or east of Svalbard and a high-pressure system prevails over Greenland (Knudsen et al. 2018), which also causes marine cold air outbreaks (Kolstad 2017). The corresponding references will be added to the paper.

**Minor concerns:**

P6L146-147: Is 33% double of 25%?
The size of the wind direction sectors is 60° (-45° to 15°) for GRL and 30° (30° to 60°) for WNB. This corresponds to 16.67% and 8.33% of 360°, respectively, which is roughly half of the observed frequency of occurrence of 33% and 16%. The sentence will be rephrased for clarity.

P7L69: Fig. 4e, f –> Fig.4c, f,
Will be corrected.

P7L171: not true

The sentence will be rephrased for clarity:

"For humidity, trends in the Odden region are up to 0.07 g kg$^{-1}$ dec$^{-1}$ larger for all other wind directions than for off-ice flow indicating that the largest humidity trends are not related to sea ice cover changes north of this region."

P7L175-176: GRL-ATM box is not in the place with the largest difference (Fig. 3 e, f)

If you refer to the WNB-ATM box, then, yes, the box should indeed be located slightly further to the north-east. Following the suggestions by referee 1, we will restructure the paper by starting with a general analysis of trends in the Fram Strait region. Based on this we will then describe in more detail how the specific study regions WNB and GRL and the placement of the ICE and ATM boxes were chosen. This will help clarify the general storyline.

P9 L180-190: These two paragraphs seem to compare the trends of air temperatures and specific humidity in WNB and GRL, respectively. If so, the sentence in L189-190 should not be placed there.
The paragraphs discuss trends of all three considered atmospheric variables in the WNB (first paragraph) and GRL (second paragraph) regions. For WNB, we start with results for temperature and humidity, followed by wind speed in L182-183: "The wind speed change is close to zero and not significant at the 95 %-level for both reanalyses." This structure is mirrored in the second paragraph for GRL and thus we will keep the sentence in L189-190 concerning wind speed trends in the GRL region.

P11L211: Typo, extent à extend
Will be corrected.

P11L214-215: "2022 was an exceptional year" Why? Explanations are only given in the discussion not here.

We rephrased this sentence to emphasize that 2022 had two periods with very low WNB fetch values, which has not been observed in previous years:

"Even though average sea ice concentrations in 2022 were not even among the 5 lowest observed years, WNB fetch values exceeded 830 km during four days in January and two days in March 2022."

For the discussion of why these periods occurred, please see our answer to your last comment (P17 L293-295).

P13L232: Three grid points are given in the reference, why the authors only show two of them?

Tetzlaff et al. (2014) presented ERA-Interim air temperatures as a function of the WNB fetch length for three points with increasing distance from the ice edge. They showed that Spearman rank correlations gradually decreased with increasing distance.  Here, we replicate their method only to demonstrate that air temperatures do not only correlate with the WNB fetch length but also with the average WNB sea ice concentration and that these correlations also decrease with increasing distance from the ice edge. To increase the clarity

of Figure 9 we only show the northernmost and southernmost grid points here. To clarify this, we write now:

"For this purpose, we use a method similar to the one by Tetzlaff et al. (2014) who ..."

P15 L256: cannot see it in Fig. 10

We will add the approximate coordinates of the region with lower correlations at the west coast of Svalbard (78.5° N, 10° E) for clarity.

P17 L280-281: Repetitive info for WNB. Maybe the authors meant "for GRL", modify the sentence accordingly

The sentence states "As for WNB" - meaning that results for GRL are similar to those from WNB. We will rewrite this sentence to:

"This is almost one third larger than trends calculated using periods with all other wind directions, which is a similar result as for WNB."

P17 L293-295: Explanations are given here for the extreme year of 2022. Justifications or references should be given.

The corresponding lines in the paper state that:
*"2022 was an extreme year since the open water fetch exceeded 800 km during two separate weeks. The event in March was a consequence of a period with strong southerly winds causing a warm-air intrusion into the Arctic across the Fram Strait region."*

During the event in March 2022 the HALO-(AC)³ aircraft campaign took place in the Fram Strait region. A paper describing the synoptic conditions during this period is currently in preparation (Walbröl et al., in prep.) and will be added as a reference. It identifies two distinct periods of warm-air intrusions on 12./13. March and 15./16. March causing advection of warm and moist air as far north as the Central Arctic Ocean (close to the North Pole).

This effect can also be seen from the following maps of temperature, humidity and wind vectors from ERA5 for a day during this period:

[Figure]

Figure: Contours of air temperature (left) and specific humidity (right) at about 10 m height from ERA5 reanalysis data on 13 March 2022. Arrows denote wind vectors at the same height. The black line denotes the 80%-sea ice concentration contour from the same day.

Such maps could be added as an appendix or supplement to the paper.

**References:**

Knudsen, E. M., Heinold, B., Dahlke, S., and 14 Co-authors (2018). Meteorological conditions during the ACLOUD/PASCAL field campaign near Svalbard in early summer 2017. *Atm. Chem. Phys.*, *18*(24), 17995-18022, https://doi.org/10.5194/acp-18-17995-2018

Kolstad, E. (2017): Higher ocean wind speeds during marine cold air outbreaks, *Q. J. Roy. Meteor. Soc.*, 143, 2084–2092, https://doi.org/10.1002/qj.3068

Walbröl, A. et al.: Meteorological overview of the HALO-(AC)³ campaign, in preparation

---

## Author Response (AR1)

Dear referee,

thank you for your comprehensive review and helpful suggestions. Since your general comments contain many points dealing with similar issues, we decided to re-group some of them by specific topics instead of providing point-by-point answers to each of them. If not stated otherwise (and with the exception of text copied directly from the new manuscript), all references to figures refer to the new figures in this document and not to the figures in the original paper draft.

1) Your first issue concerns a better explanation of the general idea of our method and the separation of flow directions. Specific suggestions were to include a schematic and to start with a more general presentation of trends by using e.g. wind roses.

We think that these are very helpful suggestions and now include a new results section (new *3.1 Trends in the Fram Strait region*) concerning general trends in the Fram Strait region.

- Starting with the map of sea ice concentration trends (original Fig. 6), we identify regions with the largest sea ice decline and define the specific study regions WNB and GRL and the corresponding ICE boxes used in later sections. We also moved the analysis of the sea ice trends in the ICE boxes (old Fig. 5 and Tab. 2) to this section.
- Afterwards, we present trend maps (without any dependence on wind direction) for temperature, humidity and wind speed for both ERA5 and MERRA-2 data (see Fig. 1)

[Figure]

Figure 1: Trend maps for atmospheric variables in the Fram Strait region based on ERA5 (left) and MERRA-2 (right) reanalysis data from January to March of the years 1992 to 2022. Trends are shown for air temperature (a,b), specific humidity (c,d) and wind speed (e,f). Dotted areas are significant at the 95%-level. Blue lines denote the 80% SSM/I-ASI sea ice concentration contours averaged from January to March for two years with large and small SIE, respectively.

- We then write: *"As a next step, we analyze wind roses for selected sub-regions, focusing especially on those regions where observed trends for sea ice cover and air temperature and humidity were largest - namely the WNB and GRL regions. We calculate the wind roses for smaller sub-regions, which we refer to as ATM boxes. The exact placement of these boxes is somewhat arbitrary and could be based on many different factors. We decided to choose locations where observed temperature and humidity trends in Fig. 1 are large. Furthermore, we choose areas that were only covered by sea ice in years with a large ice extent but were mostly ice-free in years with a small sea ice cover. Due to the large change in local ice cover we also expect to see a notable impact on atmospheric conditions in these areas "*
- We followed your suggestion and calculated wind roses for the ATM boxes. Examples for WNB are presented in Fig. 2:

[Figure]

Figure 2: Wind roses with corresponding average air temperature and specific humidity for the first (a,e) and last (b,f) ten years of the study period based on ERA5 data in the WNB ATM box. Panels (c) and (g) show the corresponding trends over the whole 31 years. Blue and orange lines denote the sectors considered for off- and on-ice flow. Maps illustrating the locations of the WNB ATM box and the flow directions for off- and on-ice flow are shown with trends for all wind directions of temperature (d) and specific humidity (h) as background shading. Blue lines denote the 80% SSM/I-ASI sea ice concentration contours averaged from January to March for two years with large (1996) and small (2018) SIE, respectively.

- In Sect. 3.2 we then present atmospheric trends for different flow directions – namely off-ice flow related to cold-air outbreaks and on-ice flow related to warm-air intrusions. Wind sectors are now chosen according to the following criteria:
  *"In the WNB ATM box, all air masses flowing from west, north or north-west originate from sea ice covered regions. However, we only select a sector of 45° containing those wind directions*

*for which atmospheric trends are the largest. We thus define off-ice flow as winds from 30° to 75° (north-easterly winds) for WNB and from -15° to 30° (northerly winds) for GRL. The obvious choice to define on-ice winds is then the sector directly opposite of the off-ice sectors. However, since the resulting sectors only contain a small fraction of less than 10% of all wind directions, we decided to widen the on-ice sector to 90°. On-ice flow is then defined as winds from 165° to 255° (south to south-westerly winds) for WNB and from 120° to 210° (south to south-easterly winds) for GRL."*

A sketch illustrating the off- and on-ice flow directions is presented in Fig. 2d for WNB, which hopefully helps better illustrate the concept of off- and on-ice flow.

- The results for trends and correlations for different flow directions, which were presented in the old Sect. 3.1 and 3.3, are then reprocessed using this new definition of off- and on-ice flow. In the GRL region, this results in only small changes (original Fig. 4 and 11). In the WBN region, changes for temperature are notable but do not change the overall conclusions (Fig. 3 e). The largest changes occur for humidity trends in the WNB region. With the new definition of wind direction sectors, humidity trends are now larger for on-ice flow in all regions (Fig. 3b,e,f). However, there are hardly any changes in the correlation maps (Fig. 4), which look almost identical for temperature and humidity.

[Figure]

Figure 3: Trend maps for ERA5 air temperature (a,c) and specific humidity (b,d) for January to March of the years 1992 to 2022 using only periods with off-ice (a,b) and on-ice winds (c,d) in the WNB region. Blue lines denote the 80% SSM/I-ASI sea ice concentration contours averaged from January to March for two years with large and small SIE, respectively. Panel (e) shows the differences of air temperature trends for off- and on-ice winds (panel (a) minus panel (c)) and panel (f) shows the corresponding trend differences for specific humidity. Dotted areas denote trends and differences of trends that are significant at the 95%-level (dark green). For trend differences, light green dots additionally mark the 90% significance level.

[Figure]

Figure 4: Differences of correlation coefficients between sea ice concentration and (e) temperature and (f) humidity trends (off-ice minus on-ice flow). White dotted areas are significant at the 95 %-level. Blue and orange lines denote the 80 % SSM/I-ASI sea ice concentration contours averaged from January to March for two years with large and small SIE, respectively. Dotted areas denote correlations and differences of correlations that are significant at the 95%-level (dark green). Light green dots additionally mark the 90% significance level.

2) The second issue concerns a generally better explanation of the methodology. What are the implications of the differences between off-ice and on-ice flow?  Why is upstream sea ice cover chosen?

Generally, we now address these issues by slightly adjusting the storyline and the overarching questions of the paper. This different framing of the topic helps to clarify the applied methods. The new structure is derived from the following questions:

1. What are the trends of near surface atmospheric variables in the Fram Strait region? (3.1)

2. Two typical synoptic situations in this region are cold air outbreaks (CAOs) with cold, dry air flowing from sea ice covered regions towards the open ocean and warm air intrusions bringing in warm and moist air during southerly flows. What atmospheric trends can we observe for these two situations – off-ice flow during CAOs and on-ice flow during warm air intrusions? (3.2)

3. One factor influencing the strength of CAOs in this region is the open water fetch or the fraction of ocean covered by sea ice. Can we find a relationship between the atmospheric trends over the open ocean in the Fram Strait region and a decrease of sea ice in the upstream region? We address this question by calculating correlations between sea ice cover and atmospheric variables for both on-ice and off-ice flow. During off-ice flow, a decreased sea ice cover means larger fluxes of heat and moisture towards the atmosphere. We would expect negative correlations in this case. During on-ice flow, winds influence the sea ice drift and push the ice edge further to the north/north-east. Furthermore, warm air masses can cause a melting of sea ice, also reducing the sea ice cover. This would also lead to negative correlations. In regions where negative correlations are stronger for off-ice flow than for on-ice flow the first mechanism likely dominates and vice versa. (3.3)

We changed the introduction accordingly and also include this information in short paragraphs at the beginning of each section in the results chapter.

2a) How can the effect of sea ice cover be entangled from large-scale atmospheric warming or increased ocean heat?

We address this issue by adding a separate new section (3.4 Discussion of mechanisms responsible for atmospheric changes):

*"Our results hint to an important role of sea ice conditions for atmospheric warming and moistening in certain areas. However, with our present analysis we cannot exclude further possible explanations for the observed atmospheric trends. For example, cases during which warm air is advected first to the north and is flowing back to the south afterwards might impact on our correlations. We mentioned already that large scale effects (especially during on-ice flow) influence the correlations. Besides the effect due to warmer air masses from the south, for example, Wickström et al. (2020) found an increase of the winter cyclone occurrence around Svalbard from 1979 to 2016. The presence of cyclones in this region was accompanied by positive temperature anomalies east of Svalbard. Such large scale synoptic changes certainly influence the general trends presented in Sect. 3.1, but not the analyses in Sect. 3.2 and 3.3, where results are separated by wind sectors. Warming due to large scale advection would result in much smoother and more homogeneous trend maps for the different wind direction sectors than we showed here in several figures. For example, it is evident from Fig. 8c and d for on-ice flow that trends are even larger north of Svalbard than further south, which would not be the case if trends were purely caused by trends within the inflow air masses or by altered cyclone tracks.*

*Our analysis also does not consider changes in sensible and latent heat transport over the ocean due to modified water temperatures that cannot be explained by sea ice decrease. The underlying processes governing the variability of ocean heat flux are very complex and not yet accurately quantified (Carmack et al., 2015; Peterson et al., 2017). Nilsen et al.*
*(2021) found that while long-term trends in the heat transport into the Arctic Ocean could be dominated by variations in the temperature of the Atlantic Water inflow, changes on shorter time-scales are more likely due to air-ocean interaction dynamics. Their analysis indicates that wind stress is an important factor influencing the variability of the West Spitsbergen Current branches flowing over the Yermak Plateau and thus the overall heat exchange in this region.*

*Since the focus of this study is mainly on the impact of sea ice changes on the atmosphere, we do not disentangle whether those sea ice changes were caused by changing wind directions, increased melting by a warmer ocean or by other factors. The sea ice cover in our two study regions has changed drastically over the last 30 years resulting in a much larger area where the atmosphere comes into direct contact with the ocean without an insulating sea ice layer. Here, we assume that an increase of the open ocean area will have a much larger impact on air temperature and humidity than an increase of the ocean temperature of a few degrees in these specific regions and thus we do not consider the latter effect explicitly. We suggest that the investigation of a combined impact of ocean and sea ice conditions should be the topic of future extended work."*

**General comments**

As I understand it, the main objective of the study is to distinguish impacts on atmospheric temperature and humidity due to "regional sea ice changes" vs "other factors influencing atmospheric conditions". The separation between the two is done by looking separately at periods of "on-ice" and "off-ice" winds, defined by specific wind direction ranges. I was occasionally confused by this approach; it seems to me that much more space needs to be given to explaining the idea behind this method and what the implications of the observed differences actually are.

See 2).

From what I understand, the authors look to separate temperature/humidity trends due to "effects of changing sea ice cover" vs "other factors" by which they seem to imply warm intrusions from the south. It is not obvious to me that the "on/off-wind" separation is a good way to do this (wouldn't it be easier to single out southerly winds, for example, or to look at heat transports directly?). Decreased regional sea ice cover presumably does not require "off-ice flow" to affect air temperature/humidity near the ice edge – I am sure the authors are aware of this, but I believe that they should lay out the motivation behind
their methodology much more clearly. Moreover, it is not clear to me that the "off-ice" direction necessarily corresponds well to actual off-ice flow.

We adjusted the off-ice wind sectors and contrast them now with southerly on-ice flows. See 1) for details.

The manuscript lacks a thorough discussion of confounding variables, and I often had trouble with the inferred causality. For example, take the statement of L309: ".. correlations for off-ice flow exceed -0.8 for both temperature and humidity and this S.I.C. changes in the upstream region can explain up to two thirds of the observed [temperature and humidity] variability..". Would they get a different result if they replaced the WNB box with a box to the NW, or just used the same box as for the atmosphere? If not, what are the implications for this statement? And what about large-scale atmospheric warming or increased ocean heat; wouldn't that affect both variables? I understand that it is hard to pick apart the many interwoven mechanisms at play, but I am missing a more clear and thorough discussion of exactly what the authors have found.

A clearer discussion is now included. See 2a) for details.

I do not recommend the publication of this manuscript in its current form; my recommendation is that the paper undergoes major revisions, which I strongly suggest should include a comprehensive overhaul of the paper with the goal of making it much clearer to the reader why the particular approach was chosen, and what one should actually make of the results. I personally suspect that it might be beneficial to separate both the results and discussion into one

section dealing with general trends/correlations and another dealing with the difference between on-ice/off-ice winds (although that would certainly not be the only way to go about it).

We now start with a presentation of general trends, followed by a comparison of cases with off-ice and on-ice flow. See 1) for details.

I recognize that the mechanism the authors invoke is somewhat complex, and that I may be missing important aspects out of ignorance. If that is the case, I hope the authors take my input as motivation for providing a clearer framing of the study in a future version of the manuscript.

**Specific comments**

- The authors need to explain more clearly how their methodology of looking at trends/correlations during different wind directions relates to main objective of the study (separating effects of regional sea ice loss vs "other effects"/southerly heat transport). Perhaps some sort of schematic could be helpful?

The method is now explained in more detail (see 2) for details). A schematic is now included (see Fig. 2d and 3).

- The study relies on the separation between "on-ice" vs "off-ice" winds. These are defined as specific wind direction ranges for the two regions. In my opinion, this choice needs to at least be justified more clearly. For example, would not winds from the NW be more directly "off-ice" at the WNB ATM box than those from the NE? And area the "WNB ICE" box and "fetch line" actually upwind of the "WNB ATM" box during off-ice winds per this definition?
It is possible that all of this would be more obvious to a reader more intimately familiar with the region than I am. However, I think it might be helpful in this respect to show some context at the start of the paper; e.g., wind roses with temperature and/or humidity, distribution of temp/hum as functions of wind direction, or map plots of the mean wind/temperature fields during on/off-ice winds might help setting the stage.

Thank you for the suggestion. We think that wind roses are very helpful to present an overview of conditions during different flow directions. They are now included in the results (Fig. 2). See 1) for details – also concerning the placement of the ATM and ICE boxes.

- It should also be made explicit, or at least discussed in more detail, whether off-ice winds cause a redistribution of heat/moisture within a larger region, or whether this is a mechanism that has caused net increases in heat/moisture in the Barents/Fram Strait area. It was not clear to me from e.g. the trend/correlation difference plots (Fig 3ef, 4cf, 10ef, 11cf) whether the positive/negative regions actually balance out.

All trends for temperature and humidity are positive regardless of wind direction (see Figures 1 and 4), which opposes the hypothesis of a redistribution of heat and moisture in this region.

- I think the authors need to state more clearly whether the differences between trends during "off-ice" and "on-ice" wind conditions are actually statistically different. Does Table 2 indicate that they are not? If so, how does that impact the conclusions?

We now calculate a t-test to compare the significance of the difference of two different trends with slopes $b$ and standard errors $s_b$ using the following formula:

$$t = \frac{b_1 - b_2}{s_{b1}^2 + s_{b2}^2},$$

with $df = n_1 + n_2 - 4$ degrees of freedom. Generally, the trends in the original Tab. 2 are significantly different only at the 80%-level or not at all. We also indicated areas where trend differences are significant at the 95%-level in the corresponding maps (see Fig. 3e,f). There, also only a small fraction is significant. With the refinement of the research questions and the overall structure of the paper described in 1) and 2) above, however, it is not the main focus of this section – where we analyze trends for on-ice and off-ice flow – to detect whether trends are significantly different. The aim is rather to compare the magnitude of the trends. It is then the goal of the following section to investigate the impact of the sea ice cover on atmospheric conditions. Our analysis of correlation differences already included a significance test.

- Throughout the manuscript, there needs to be a clearer differentiation between trends/effects that are attributed specifically to "off-ice flow" vs to other effects/"general trends".
One specific example: From L5 in the abstract ("During off-ice flow.."): It seems necessary here to include the corresponding temperature changes during the other wind directions.
Another example: Red markers in Figure 9 show the relationship between air temperature NW of Svalbard as a function of WNB S.I.C./polynya length. How different would this figure look if you only included "on-ice" winds?

We now describe the differences between on-ice and off-ice flow in more detail. See also 1) and 2).

We know include Fig. 5 in the Appendix, which is similar to Fig. 9 in the original manuscript, but this time also includes off-ice flow. Panel b shows the more northern of the original two points. We write now:

*"It is evident from Fig 5 that it highly depends on the specific region whether correlations for off- and on-ice flow differ from each other. For a point located at the north-west corner of Svalbard (Fig 5b) correlation coefficients are almost identical, while off-ice correlations are larger than those for on-ice flow in a region slightly further toward the west (Fig.5a)."*

Generally, the shape change of temperature with sea ice concentration is very similar for both on-ice and off-ice flow and also correlation values do not differ much. It is evident from the

correlation maps in Fig. 4 correlation differences are highly variable depending on the region.

[Figure]

Figure 5: ERA5 air temperature averaged monthly for periods with off-ice (orange) and on-ice (green) flow as function of average sea ice concentrations in the WNB ICE box. Numbers are Spearman rank correlations.

- L129: The formula by Steiger 1980, or a brief description of what it is, should be included.

The following description is now included in Sect. 2.4:

*"To test whether correlation coefficients differ significantly from each other, we use the formula for independent correlation coefficients (Steiger, 1980), which is based on Fisher's z-transform:*

$$z = \frac{1}{2} \ln \left( \frac{1+r}{1-r} \right).$$

*A t-test is then applied to the difference of the z-values."*

- Figure A1: I am a little confused as to why the SD of sea ice concentration is shown here. Why not just show the actual (winter average) concentration?

In our opinion, the outline of the Odden ice tongue can be visually detected more clearly from the standard deviations than from the mean sea ice concentration. This can be seen from Fig. 6. Nevertheless, we now include maps of the means as an additional figure in the appendix.

[Figure]

Figure 6: Mean (top) and standard deviation (bottom) of January to March SSM/I-ASI sea ice concentrations.

- Great that the study looks at two different reanalysis products, this strengthens the analysis. Figure 5 shows apparent striking systematic differences in both temperature and humidity – it would be useful if the authors could briefly comment on possible reasons for this (different height levels? known biases?).

We use MERRA-2 data provided at 10 m height and ERA5 from the lowest model level, which varies roughly from 8 to 10 m height in this region and season. Such a small height difference between the two reanalyses does not explain the observed differences. A few studies have focused on the analysis of systematic differences between reanalyses in this region. We now briefly discuss their findings at the end of Sect. 3.2.:

*"A few previous studies focusing on the analysis of systematic differences between reanalyses in this region have provided similar results. Graham et al. (2019) compared reanalysis data to meteorological observations during the N-ICE campaign in winter 2015 and found that ERA5 has a stronger overestimation of winter temperatures, while MERRA-2 has a stronger overestimation of*

*the downward longwave radiation flux. Comparing reanalyses with remote sensing data also revealed that MERRA-2 overestimates the near-surface cloud fraction, which increases longwave cloud radiative effects (Yeo et al., 2022). More research is necessary to assess the underlying causes for these biases in more details."*

In general, I found the figures to be nicely made and helpful. I would suggest a few modifications:

- Add scale bars to the maps (helps to interpret statements like "500 km downstream", etc).

The maps now include scale bars (see Fig. 1 and 3).

- Clearly label the boxes – e.g. GRB (ICE), GRB (ATM) or similar; it is at times difficult to follow which is which. Label the Odden Ice Tongue somewhere in Figure 1. The Odden ice tongue should also be indicated in Fig. 4.

We updated the region map and labeled the boxes and regions more clearly (see Fig. 7).

[Figure]

Figure 7: Overview maps of the study region. (a) January to March averages of the SSM/I-ASI sea ice concentration are displayed as grey contours from 15 % upward in 5%-steps for 1996 (a year with large SIE) and as orange 15%-contour for 2018 (a year with small SIE). The 15%-contour of the maximum sea ice concentration in 1998 (a year with a pronounced Odden ice tongue) is shown as dashed line. (b) Location of the specific study areas: Large ICE boxes indicate the areas considered for trends of sea ice conditions and small ATM boxes indicate areas considered for atmospheric trends in the Greenland Sea (GRL, blue) and the Western Nansen Basin (WNB, red) regions, respectively. The thick green line indicates the path considered for WNB fetch calculations.

- Revise the colormaps such that warm colors correspond to warm temperatures etc (this would make especially Figures 3/4 a bit more intuitive).

We now omit the purple end of the colormap for temperature trends so that red colors correspond to regions with the largest warming (see Fig. 1 and 3).

- Figures 3/10 and 4/11 should at least have the same color showing the same sign of temp/hum change.

The original Fig. 10 and 11 do not show trends but correlation coefficients. We originally chose to use the same color maps as in the original Fig. 3 and 4 to make it easy to distinguish results for temperature and humidity. However, we see that this might be misleading and now use a different colormap for all correlation plots (original Fig. 10 and 11 a-d). An example is shown in Fig. 8.

[Figure]

Figure 8: Spearman rank correlation coefficients between monthly WNB sea ice concentrations and air temperatures (a) and specific humidity (b) from ERA5 averaged for periods with off-ice flow. Blue and orange lines denote the 80% SSM/I-ASI sea ice concentration contours averaged from January to March for two years with large and small SIE, respectively.

- The area NW of Svalbard where trends are most affected by off-ice winds (e.g. Fig. 3ab, Fig3ef) seems to correspond roughly to the Yermak Plateau, which from what I understand is an area where the upper ocean is particularly warm and loses a lot of heat and moisture to the atmosphere. Could this play a role in the mechanism that the authors invoke? (Note: I don't expect the authors to go into detail, but I think it warrants a mention).

With the updated definition of wind direction sectors the original Fig. 3 has changed a bit in this specific region (see Fig. 3) so that positive trends do not extend as far to the north-west as before. Nevertheless, we checked whether an impact of the Yermak plateau is notable in the correlation maps (see Fig. 9). It is evident that the area of maximum correlation differences for on-ice and off-ice flow do not align very well with the outline of the Yermak plateau and thus it is unlikely that local heat exchange due to ocean mixing dominates the observed trends for off-ice flow.

Nevertheless, we mention the general importance of the Yermak plateau on the ocean circulation in this region in the discussion. See also 2) for details.

[Figure]

Figure 9: Same as Fig. 5a but with an added isoline at 1500m depth to illustrate the location of the Yermak plateau. Based on GEBCO bathymetry data.

**Technical/minor**

- L20-22: Meaning is unclear.

The sentence has been rephrased:

*"The strength of the trend also highly depends on the region. Stroeve and Notz (2018) and Onarheim et al. (2018) found that the regions with the largest decrease of SIE were the Beaufort and East Siberian Seas in September and in the Barents and Greenland seas in March."*

- L147 and onward: "trends of the frequencies" – I find this use of "frequency" confusing (others may not)

What we mean here is the *"frequency of occurrence"*. This term is now used for clarity.

- L142. "Westerly to northerly": If this refers to the 30-60 degree window, this phrasing seems inaccurate.

This sentence referred to the -45° to 15° window. It is no longer included due to the restructuring of the manuscript and the updated definitions of the wind sectors.

- L173: "time series of trends" – meaning unclear

It should read *"time series of atmospheric variables"* and is now corrected.

- I would advise being careful with the use of "as for"; to me, this reads as "with regard to". (Ex. L189, L245, L280).

All respective sentences have been rephrased, e.g.:

L189: *"Like for WNB, …"*

L245: *"Similar to the atmospheric trends (Sect. 3.1), the general patterns …"*

**References**

Carmack, E., Polyakov, I., Padman, L., Fer, I., Hunke, E., Hutchings, J.,et al. (2015). Toward quantifying the increasing role of oceanic heat in sea ice loss in the new Arctic. Bulletin of the American Meteorological Society, 96(12), 2079-2105, 10.1175/BAMS-D-13-00177.1

Graham, R. M., Cohen, L., Ritzhaupt, N., Segger, B., Graversen, R. G., Rinke, A., Walden, V. P., Granskog, M. A., and Hudson, S. R. (2019): Evaluation of Six Atmospheric Reanalyses over Arctic Sea Ice from Winter to Early Summer, *Journal of Climate*, *32*(14), 4121-4143, 10.1175/JCLI-D-18-0643.1

Nilsen, F., Ersdal, E. A., & Skogseth, R. (2021). Wind-driven variability in the Spitsbergen Polar Current and the Svalbard Branch across the Yermak Plateau. Journal of Geophysical Research: Oceans, 126, e2020JC016734, 10.1029/2020JC016734

Peterson, A. K., Fer, I., McPhee, M. G., & Randelhoff, A. (2017). Turbulent heat and momentum fluxes in the upper ocean under Arctic sea ice. Journal of Geophysical Research: Oceans, 122(2), 1439-1456, 10.1002/2016JC01228

Wickström, S., Jonassen, M. O., Vihma, T., & Uotila, P. (2020). Trends in cyclones in the high-latitude North Atlantic during 1979–2016. Quarterly Journal of the Royal Meteorological Society, 146(727), 762-779, 10.1002/qj.3707

Yeo, H., Kim, M. H., Son, S. W., Jeong, J. H., Yoon, J. H., Kim, B. M., & Kim, S. W. (2022). Arctic cloud properties and associated radiative effects in the three newer reanalysis datasets (ERA5, MERRA-2, JRA-55): Discrepancies and possible causes. Atmospheric Research, 270, 106080, 10.1016/j.atmosres.2022.106080

Dear referee,

Thank you for your helpful comments. Please find point-by-point answers below.

1) How could the effects of ocean can be separated from that due to pure ice changes?

We added a separate new section (3.4 Discussion of mechanisms responsible for atmospheric changes) in which we also discuss possible ocean impacts:

*"Our analysis also does not consider changes in sensible and latent heat transport over the ocean due to modified water temperatures that cannot be explained by sea ice decrease. The underlying processes governing the variability of ocean heat flux are very complex and not yet accurately quantified (Carmack et al., 2015; Peterson et al., 2017). Nilsen et al. (2021) found that while long-term trends in the heat transport into the Arctic Ocean could be dominated by variations in the temperature of the Atlantic Water inflow, changes on shorter time-scales are more likely due to air-ocean interaction dynamics. Their analysis indicates that wind stress is an important factor influencing the variability of the West Spitsbergen Current branches flowing over the Yermak Plateau and thus the overall heat exchange in this region.*

*Since the focus of this study is mainly on the impact of sea ice changes on the atmosphere, we do not disentangle whether those sea ice changes were caused by changing wind directions, increased melting by a warmer ocean or by other factors.  The sea ice cover in our two study regions has changed drastically over the last 30 years resulting in a much larger area where the atmosphere comes into direct contact with the ocean without an insulating sea ice layer. Here, we assume that an increase of the open ocean area will have a much larger impact on air temperature and humidity than an increase of the ocean temperature of a few degrees in these specific regions and thus we do not consider the latter effect explicitly. We suggest that the investigation of a combined impact of ocean and sea ice conditions should be the topic of future extended work."*

2) Would you mind considering using a model to confirm the main conclusions of this study?

We agree, modeling would strengthen the conclusions, but this is beyond the scope and possibilities of this paper. It would require a very comprehensive analysis. In fact, one of the goals of this paper is to stimulate future modeling work based on our present results. We added this idea to the outlook paragraph at the end of the section *4. Summary and conclusions*:

*"We also suggest conducting modeling studies in future research to strengthen the conclusions found in this paper."*

3) Do the off-ice events have associations with typical large-scale atmospheric circulations? Or is there any connection with synoptic cyclones?

Events of off-ice flow in the Fram Strait region typically occur when a low-pressure system is present over or east of Svalbard and a high-pressure system prevails over Greenland, which

also causes marine cold air outbreaks. The corresponding references are added to the paper:

"During off-ice flow, cold and dry air masses from sea ice covered regions (e.g. the Central Arctic) flow towards the open ocean. These so-called cold air outbreaks (CAOs) typically occur in the Fram Strait region when a low-pressure system is present over or east of Svalbard and a high-pressure system prevails over Greenland (Knudsen et al., 2018; Kolstad, 2017})."

**Minor concerns:**

P6L146-147: Is 33% double of 25%?
The definition of the wind sectors has been changed (following the suggestions of the first referee) and thus the sentence had to be rephrased. The new off-ice sectors span 45° and thus we would expect a wind direction frequency of 12.5% for a uniform wind distribution:

*"Off-ice flow conditions are present for about 21% of the time in the WNB region and for about 40% in the GRL region, which is much larger than what would be expected for a uniform distribution of wind directions (12.5%)."*

P7L69: Fig. 4e, f –> Fig.4c, f,
Corrected

P7L171: not true

Due to the updated definition of the wind direction sectors for the different flow directions, this sentence is not included in the manuscript anymore.

P7L175-176: GRL-ATM box is not in the place with the largest difference (Fig. 3 e, f)

We slightly adjusted the placement of the ATM boxes. Following the suggestions by referee 1, we restructured the paper by starting with a general analysis of trends in the Fram Strait region. Based on this we then describe in more detail how the specific study regions WNB and GRL and the placement of the ICE and ATM boxes were chosen:

*"The exact placement of the ATM boxes is somewhat arbitrary and could be based on many different factors. We decided to choose locations where observed temperature and humidity trends are large. Furthermore, we choose areas that were only covered by sea ice in years with a large ice extent but were mostly ice-free in years with a small sea ice cover. Due to the large change in local ice cover we also expect to see a notable impact on atmospheric conditions in these areas."*

P9 L180-190: These two paragraphs seem to compare the trends of air temperatures and specific humidity in WNB and GRL, respectively. If so, the sentence in L189-190 should not be placed there.
The paragraphs discuss trends of all three considered atmospheric variables in the WNB (first paragraph) and GRL (second paragraph) regions. For WNB, we start with results for temperature and humidity, followed by wind speed in L182-183: "The wind speed change is close to zero and not significant at the 95 %-level for both reanalyses." This structure is

mirrored in the second paragraph for GRL and thus we will keep the sentence in L189-190 concerning wind speed trends in the GRL region. The paragraph has been rephrased to:

*"In the WNB ATM box, air temperatures and specific humidity show significant positive trends with slightly smaller numbers for MERRA-2 than for ERA5 for off-ice flow. During the 31-year period, air temperatures increased by up to 10K and the humidity by 0.7g/kg. The wind speed change is close to zero and not significant at the 95%-level for both reanalyses. For on-ice flow, trends for temperature are up to 50% smaller, while trends for humidity are up to 25% larger than for off-ice flow."*

P11L211: Typo, extent à extend
Corrected

P11L214-215: "2022 was an exceptional year" Why? Explanations are only given in the discussion not here.

We rephrased this sentence to emphasize that 2022 had two periods with very low WNB fetch values, which has not been observed in previous years. We also moved the explanation from the discussion to this section:

*"Even though average sea ice concentrations in 2022 were not even among the five lowest observed years, WNB fetch values exceeded 830km during four days in January and two days in March 2022. The event in March was a consequence of a period with strong southerly winds causing a warm-air intrusion into the Arctic across the Fram Strait region (Walbröl et al.,2023). See Fig. C1 for an example of the flow conditions on 13 March 2022. The only other period with such high WNB fetch values was during the first days of January 2013, however, the sea ice cover quickly closed up during the next weeks in this year."*

P13L232: Three grid points are given in the reference, why the authors only show two of them?

Tetzlaff et al. (2014) presented ERA-Interim air temperatures as a function of the WNB fetch length for three points with increasing distance from the ice edge. They showed that Spearman rank correlations gradually decreased with increasing distance.  Here, we replicate their method only to demonstrate that air temperatures do not only correlate with the WNB fetch length but also with the average WNB sea ice concentration and that these correlations also decrease with increasing distance from the ice edge. To increase the clarity of Figure 9 we only show the northernmost and southernmost grid points here. To clarify this, we write now:

*"We extend their analysis to the year 2022 and use a higher temporal resolution and consider January to March monthly means instead of winter means. Exemplarily, we show data from two points with locations close to P1 and P3 in Tetzlaff et al. (2014)."*

P15 L256: cannot see it in Fig. 10

We added the approximate coordinates of the region with lower correlations at the west coast of Svalbard (78.5° N, 10° E) for clarity.

P17 L280-281: Repetitive info for WNB. Maybe the authors meant "for GRL", modify the sentence accordingly

We meant the following: "This is almost one third larger than trends calculated using periods with all other wind directions, which is a similar result as for WNB." However, due to the updated definition of the wind direction sectors for the different flow directions, this sentence is not included in the manuscript anymore.

P17 L293-295: Explanations are given here for the extreme year of 2022. Justifications or references should be given.

The corresponding lines in the paper stated that:
"2022 was an extreme year since the open water fetch exceeded 800 km during two separate weeks. The event in March was a consequence of a period with strong southerly winds causing a warm-air intrusion into the Arctic across the Fram Strait region."

This effect can also be seen from the following maps of temperature, humidity and wind vectors from ERA5 for a day during this period:

[Figure]

Figure C1: Contours of air temperature (a) and specific humidity (b) at about 10 m height from ERA5 reanalysis data on 13 March 2022. Arrows represent the corresponding wind vectors. The black line denotes the 80%-sea ice concentration contour from the same day.

The figure and a description are added in the appendix:

"Appendix C: Synoptic conditions in 2022

2022 was a year with two periods with very low sea ice concentration in the Whaler's Bay polynya - one at the end of January and one in the middle of March (see Fig. 12). During the event in March 2022 the HALO-(AC)³ aircraft campaign took place in the Fram Strait region. A paper describing the synoptic conditions during this period is currently under review (Walbröl et al., 2023). It identifies two distinct periods of warm-air intrusions on 12./13. March and 15./16. March causing advection of warm and moist air as far north as the Central Arctic Ocean (close to the North Pole). This effect can also be seen from the maps of temperature, humidity and wind vectors from ERA5 for a day during this period presented in Fig. C1. Southerly winds push the sea ice northward and thus are one of the factors causing a large open water area north of Svalbard."

**References:**

Knudsen, E. M., Heinold, B., Dahlke, S., and 14 Co-authors (2018). Meteorological conditions during the ACLOUD/PASCAL field campaign near Svalbard in early summer 2017. *Atm. Chem. Phys.*, *18*(24), 17995-18022, https://doi.org/10.5194/acp-18-17995-2018

Kolstad, E. (2017): Higher ocean wind speeds during marine cold air outbreaks, *Q. J. Roy. Meteor. Soc.*, 143, 2084–2092, https://doi.org/10.1002/qj.3068

Walbröl, A., Michaelis, J., Becker, S., Dorff, H., Gorodetskaya, I., Kirbus, B., Lauer, M., Maherndl, N., Maturilli, M., Mayer, J., Müller, H., Neggers, R. A. J., Paulus, F. M., Röttenbacher, J., Rückert, J. E., Schirmacher, I., Slättberg, N., Ehrlich, A., Wendisch, M., and Crewell, S. (2023): Environmental conditions in the North Atlantic sector of the Arctic during the HALO–(AC)3 campaign, EGUsphere [preprint], 2023, 1–48, https://doi.org/10.5194/egusphere-2023-668

---

## Author Response (AR2)

We thank the two anonymous reviewers and the editor for the time and effort they invested in critically reviewing our manuscript. Please find answers to your very helpful comments and suggestions below.

**Report #1**

1) In the abstract, for example, the words "off-ice flow" and "on-ice flow" suddenly appeared. I could not understand at first what these were, and also, I could not understand why the authors made this kind of analysis (I found them after I read on).

We rephrased the respective sentence in the abstract for clarity: *"As a next step, two typical flow directions for this region were studied: cold-air outbreaks with northerly winds originating from ice covered areas (off-ice flow) and warm-air intrusions with southerly winds from open ocean regions (on-ice flow)."*

2) While I respect the authors' considerations, I still believe that in the sea-ice region, it is essential to consider the variability from the viewpoint of the coupled atmosphere-ice-ocean system. Of course, it is understandable that there are limits to the types and amount of data available in sea-ice covered areas. Even so, the authors should conduct the quantitative discussions from the viewpoint of the coupled system. As I noted earlier, the lack of these perspectives may be causing the difficulty in understanding.

Analyzing the coupled atmosphere-ice-ocean system is certainly important when studying large scale processes or to predict climate change impacts. However, we think that looking only at the isolated interaction of two components of the coupled system helps to deepen our understanding of specific processes and thus also has its value. The literature cited in our introduction also indicates that this is common in the field, with some studies focusing mainly on the relationship between sea ice cover and either atmospheric conditions (Dahlke et al., 2020; Isaksen et al., 2016) or ocean processes (Selyuzhenok et al., 2020). As the analyses in our paper are already rather extensive, we still think that including also the ocean impact should be the topic of future research, as we stated in the conclusions.

3) This study is quite regional as the authors described in "4. Summary and conclusions", as "The goal of this study was to analyze recent regional trends in near-surface atmospheric variables in the Fram Strait region in winter and their connection to regional sea ice cover". Also, my impression from reading the manuscript is that the analyses in this study are similar to those done in previous studies such as Tetz 14. In other words, I feel that this study can be regarded as a temporal extension of those by adding recent data. Of course, I can read and understand that various new findings have been obtained by adding the latest data.

Yes, our primary goal was to conduct a regional study, as we firmly believe that such studies are important alongside large-scale research. We are also not the first ones following the idea of

addressing the regional aspect, which has already been explored in numerous studies for both polar and mid-latitude regions. The results of our study are most important for the inhabitants of Svalbard, but regional change also has an impact on ship traffic, making it a matter of concern for various stakeholders.

While our study shares similarities with Tetz14, we acknowledge its influence as a basis for our work. However, our new research expands well beyond Tetz14, as explained in our response to comment 3b). We believe that our present study offers more than just the addition of new data. It sheds light on the continuous relevance of the trends identified in Tetz14, as they persist and are still relevant.

Combined with the editor's comments:

3a) There is a paragraph in the Conclusions section missing, widening your results and explaining what they mean for our understanding of Arctic sea ice decline.

We added the following paragraph to the conclusions:

*"Our study, though regional in scope, provides valuable insights into the strong connection of atmospheric and sea ice conditions. Some of the results can be transferred to other regions. Obviously, wind direction plays an important role for regional climate change near the marginal sea ice zones. This finding can most likely be generalized to all areas near the ice edge. Furthermore, our research highlights the large spatial variability of sea ice decline and its corresponding atmospheric response. The large trends near Svalbard align with previous studies that identify Svalbard as a hotspot in Arctic climate change. Our study contributes to a more comprehensive and detailed understanding of the ongoing changes in the Arctic."*

3b) You could also more specific to what extent your results confirm previous studies such as Tetzlaff et al. (2014) or provide additional knowledge.

The main results of Sect. 3.3 are based on the correlation maps, which enable us to conduct a detailed spatial analysis of patterns and to identify regions in which a substantial part of atmospheric change can be attributed to sea ice changes. The extension of the time series by Tetz14 was mostly a prerequisite in order to introduce the method used for the calculation of the maps. We try to make this clearer by adding the following sentences to the third paragraph of Sect. 3.3: *"Consequently, we analyze in this section how atmospheric changes are related to sea ice changes. We use a method inspired by the study by Tetz14, who correlated air temperatures with upstream sea ice conditions. While their analysis focused only at three locations over the open ocean with increasing distance to the ice edge, we extend their analysis not only in time, but also present maps showing correlation coefficients for each reanalysis grid point. Looking at spatial patterns instead of single points has the clear advantage that we can identify specific regions in which a substantial part of atmospheric change can be attributed to sea ice changes."*

**Report #2**

Minor comments

1) L191-197: Would probably be appropriate to add the more recent analysis by Isaksen et al 2022 (doi:10.1038/s41598-022-13568-5) to this discussion.

We now also include this paper in the discussion: *"In a recent study, Isaksen et al. (2022) presented air temperature trends based on weather station and reanalysis data averaged from December to February. Between 1991 and 2020, trends in the region north-east of Svalbard were in the order of 3 K per decade based on ERA5 data and even slightly larger values based on station data and the CARRA reanalysis, which is also in line with our findings."*

2) I was a little confused about c, g: What do the wind rose direction/amplitude (the shape, not the colours) represent in these panels? Based on the caption I was expecting them to show the *trend* in winds, but I guess it must be the 1992-2002 wind rose (?). Maybe clarify in the caption.

The first part of the caption now explains this in more detail: *"Wind roses based on ERA5 data in the WNB ATM box for the first ten years of the study period (a,e), the last ten years (b,f), and the whole period from 1992 to 2022. The length of the bars indicates the observed frequency of winds from each wind direction sector spanning 15°. The color denotes the average air temperature (a,b) and specific humidity (e,f) for the respective periods. The colors in panels (c) and (g) show the corresponding temperature and humidity trends over the whole 31 years."*

3) L216-17: "The situation..": Hard to see this in esp. Fig 6g..

The original aim was to use the same limits (0 and 0.4 g/kg/dec) for both regions. However, we agree with your point and adjusted the limits of the colormaps for Figures 5g and 6g separately. To clarify this, the following sentence was added to the caption of Fig. 6: *"Note that the colormap of panels (g) and (h) spans a smaller range than in Fig. 5."*

4) L283-285: Is this contrary to your results, then? ($T\_MERRA2 > T\_ERA5$)? If so, warrants a comment.

The respective paper found biases of winter temperatures of 3 K for MERRA-2 and 3.4 K for ERA5. Thus, you are correct, and this somewhat contradicts our findings. Compared with biases for other reanalyses (e.g. 1.1 K for JRA-55), however, the biases for ERA5 and MERRA-2 are rather similar to each other. We rephrased the sentence to reflect these points: *"Graham et al. (2019) compared reanalysis data to meteorological observations during the N-ICE campaign in winter 2015 and found that both ERA5 and MERRA-2 overestimate winter temperatures by more than 3 K. Contrary to our results, however, biases are slightly larger for ERA5 than for*

*MERRA-2. They also found that among all the reanalyses considered, MERRA-2 has the largest overestimation of the downward longwave radiation flux."*

5) L288: "atmospheric flows" -> ? Should this be replaced with e.g. "temperature and humidity"?

We now write *"atmospheric variables"*, since besides temperature and humidity we also look at wind speed in this section.

6) Figure 11/paragraph around L315: Would be good to add an axis with month names for quick reference vs the text.

An axis with month names was added to the top of the figure.

7) Somewhere around L315: Recommend quickly repeating how you calculate fetch length (open water along green line o.s.).

The following description was added: *"As described in Sect. 2.3, the WNB fetch is the distance along the green line in Fig. 1 over mostly open ocean (grid cells with a sea ice concentration below 70 %)."*

8) L327: "exemplarily" -> "for example"?

What we meant and write now is *"as an example"*.

9) Figure 12: Needs a legend and/or explanation in caption! Takes a while to figure out what blue triangles vs red circles are.

We now call the two considered locations P1 and P2 for clarity, added the legend also to panel a), and expanded the caption.

[Figure]

**Figure 12:** ERA5 air temperature at two locations (marked with triangles in Fig. 13) averaged monthly for January to March from 1992 to 2022 for periods with off-ice flow as function of WNB fetch (a) and of average sea ice concentrations in the WNB ICE box (b). P1 (red circles) is located close to the ice edge north-west of Svalbard and P2 (blue triangles) is located about 250 km further to the south-west over the open ocean region. Numbers are Spearman rank correlations. The dashed line indicates the freezing temperature of sea water

10) L331: "very similar": Similar to each other or to the polynya length correlations? Best to rephrase.

The sentence was rephrased to: *"We performed a similar analysis to determine the relationship between average WNB sea ice concentration and atmospheric temperatures and found that correlations at the two considered points are very similar to those using open-water fetch."*

11) Figure 13/14: Should make it clear what "difference" mean is (on-ice minus off-ice vs the other way around) in ylabel or caption – could help the reader by stating "red/blue colours mean that X/Y".

We added a more detailed description to the figure captions. This is the text for Figure 13: *"Panels (e) and (f) show the corresponding differences of correlation coefficients between off-ice*

*and on-ice flow (panel (a) minus panel (c) and panel (b) minus panel (d)). Red colors mean that negative correlations are larger for off-ice flow compared with on-ice flow."*

12) L346: "Over..": And winds presumably also deflect around the steep land topography?

We agree and added the following sentence: *"On the one hand, the flow likely divides around Svalbard when winds deflect around the steep land topography. On the other hand, air masses from north-east do not experience any further warming and moistening over land and thus correlations decrease."*

13) L349: It's a little ambiguous what "differences" refers to here – seems to indicate difference between temp and hum patterns (leading me to look for differences between 13e and 13f), but I don't think that is intended? Perhaps rephrase for clarity.

The sentence was rephrased for clarity: *"We also calculated the differences between the correlation maps for off- and on-ice flow for both temperature (Fig. 13e) and humidity (Fig. 13f). It is remarkable that the maps for temperature and humidity look almost identical, showing two areas of distinct differences between correlations for off- and on-ice flow."*

14) L352: typo? (86N)

It should be 83° N and has been corrected.

15) L383: "although the effect was smaller" – phrasing seems strange, maybe "smaller but significant" or similar?

The sentence was rephrased to: *"While the impact of sea ice decrease was most pronounced during off-ice flow, smaller but significant negative correlations were also found during on-ice flow."*

16) L422-L426: Look over these sentences for clarity (stray "with", and what does "which" point to).

The sentences were rephrased to: *"In the GRL region, the negative sea ice trend is also significant but relatively smaller at -4.7% per decade compared to WNB. This discrepancy can be partially attributed to a higher proportion of open ocean within the GRL area already at the beginning of the analyzed period. The first half of the examined period shows the greatest inter-annual variability, likely influenced by the presence of the Odden ice tongue in certain years before 2006."*

17) Section 4 is helpful, but I would suggest looking over it once more for clarity/language.

The respective section has been checked thoroughly and many sentences were rephrased for clarity. Please see the revised version for all improvements in this section.